# Selective control of prefrontal neural timescales by parietal cortex

Orhan Soyuhos [1,2], Marc Zirnsak[3], Rishidev Chaudhuri [1,4,5] & Xiaomo Chen [1,4] ✉

The frontal eye field (FEF) and posterior parietal cortex (PPC) are key hubs of the dorsal attention network, yet how parietal inputs shape prefrontal circuit temporal dynamics and attentional computations remains largely unknown. We measured intrinsic timescales of FEF neurons in male rhesus macaques and examined their changes during PPC inactivation. We observed two distinct classes of FEF neurons based on their intrinsic timescales: short-timescale neurons (~25 ms) and long-timescale neurons (~100 ms). Short-timescale neurons showed stronger transient visual responses, suggesting a role in rapid visual processing. In contrast, long-timescale neurons exhibited stronger sustained salience representation, suggesting a role in spatiotemporal integration to maintain stimulus-driven attention. During PPC inactivation, intrinsic timescales increased in both neuron types, with a substantially larger effect in short-timescale neurons. In addition, PPC inactivation selectively disrupted salience computation, particularly in long-timescale neurons. These findings provide causal evidence linking intrinsic local neural timescales to long-range inter-area communication and indicate the presence of at least two distinct network motifs that support different neuronal dynamics and functional computations within the FEF.

Intrinsic neural timescales measure the duration over which spontaneous neural activity remains temporally correlated with its prior states, independent of external stimuli or specific behavioral tasks[1–3]. These timescales define the temporal window in which previous neural activity can influence current neural dynamics, thereby shaping how information is processed and integrated within neural circuits[4]. Across cortical regions, intrinsic timescales exhibit systematic variations that reflect functional specialization. For instance, along the visual hierarchy, early visual areas exhibit shorter intrinsic timescales, whereas higher-order areas demonstrate longer timescales[2,5]. This gradient aligns with functional hierarchies in the brain, where shorter timescales are associated with regions involved in transient information processing, and longer timescales support sustained cognitive functions, such as working memory and decision-making[6–9]. Recent studies

further suggest that intrinsic timescales also vary among individual neurons within a single brain region, corresponding to the functional heterogeneity at the neuronal level[1,10,11]. Taken together, this diversity may critically enable the brain's dynamic balance between integrating and segregating information over multiple temporal scales[1,3,4,12,13].

In the primate brain, the frontal eye field (FEF) and the posterior parietal cortex (PPC) are key nodes in the frontoparietal attention network, with the PPC located earlier in the visual hierarchy[14–16]. Both areas are involved in stimulus-driven and goal-directed visual attention[17,18]. Consistent with this network organization, our recent research demonstrated a causal role of the PPC in regulating stimulus-driven attentional representations in the FEF and attention-driven behavior[19]. Anatomically, the FEF and oculomotor areas within the PPC, such as the lateral intraparietal area (LIP), exhibit extensive reciprocal

[1]Center for Neuroscience, University of California, Davis, CA, USA. [2]Department of Psychology, University of California, Davis, CA, USA. [3]Department of Neurobiology, Howard Hughes Medical Institute, Stanford University, Stanford, CA, USA. [4]Department of Neurobiology, Physiology and Behavior, University of California, Davis, CA, USA. [5]Department of Mathematics, University of California, Davis, CA, USA. ✉e-mail: xmch@ucdavis.edu

connectivity[20]. These interactions between FEF and PPC are thought to play a crucial role in supporting various cognitive functions, including visual attention[21–24]. Moreover, the FEF has direct anatomical connections with most visual cortical areas, both receiving input and sending feedback projections across the visual cortex[25,26]. Therefore, FEF is considered a proxy for prefrontal cortex communication with visual areas through these anatomical connections[27]. Consequently, intrinsic timescales within FEF likely play an important role in shaping attentional control across prefrontal and visual cortices.

The mechanisms underlying the variability in intrinsic neural timescales are thought to involve both inter- and intra-area interactions[28]. Previous studies have primarily emphasized the role of local and regional factors within the brain, such as dendritic spine density in pyramidal neurons[29–31], the expression levels of NMDA and GABA receptor genes[32–34], and the extent of structural and functional connectivity[22,35]. Complementing these empirical studies, computational modeling work has further shown that differences in intrinsic timescales across cortical regions can emerge naturally from variations in both local excitatory-inhibitory interactions and long-range connectivity patterns[5,28,36,37]. These findings suggest that intrinsic timescales arise from a combination of diverse excitatory and inhibitory connection strengths within regions and long-range connectivity patterns, indicating that neither factor alone is sufficient to predict a region's timescales[5,36,38]. However, despite the modeling work, a direct causal link between inter-area communication and intrinsic neural timescales remains untested. Specifically, it is entirely unknown whether the intrinsic timescales of FEF neurons depend on PPC input.

In this study, we examined the diversity of intrinsic neural timescales among FEF neurons, assessed their relationship to neuronal functional specialization during visual processing and salience representation, and investigated their dependence on PPC inputs. We found a bimodal distribution of intrinsic timescales, identifying distinct FEF neuronal groups with either fast (short-$\tau$) or slow (long-$\tau$) timescales. Moreover, these timescales measured during the baseline period correlated with the neurons' functional properties during the task: short-$\tau$ neurons were more involved in processing transient visual input, while long-$\tau$ neurons exhibited stronger sustained salience representation. Finally, PPC inactivation selectively increased the intrinsic timescales of short-$\tau$ neurons while having a larger impact on salience modulation of long-$\tau$ neurons. Together, these findings provide direct causal evidence that PPC inputs regulate intrinsic dynamics and attentional modulation within FEF, suggesting a distributed computation mechanism underlying the frontoparietal attention network.

## Results

We recorded spiking activity from the right FEF of two behaving monkeys (J and Q) using multichannel microelectrodes (Fig. 1a; see Methods). These neuronal responses were measured during the presentation of two types of visual stimuli: single visual probes and pop-out arrays containing a unique stimulus (Fig. 1b). To assess the impact of PPC input on FEF activity, we reversibly inactivated the PPC using cryoloops chronically implanted within the right intraparietal sulcus (IPS). We then compared FEF spiking responses between PPC inactivation and control conditions. In total, we recorded 400 single- and multi-unit responses from 192 FEF recording sites across 11 experimental sessions.

### Two distinct neural timescales in FEF neurons

To characterize the diversity of intrinsic neuronal dynamics within the FEF, we first analyzed neural timescales in two monkeys under the control condition (no PPC inactivation). Intrinsic neural timescales were quantified by calculating the spike-count autocorrelation for each neuron based on spontaneous activity in the baseline epoch before stimulus presentation (Fig. 1c). Autocorrelation functions were fit using a single exponential decay model to estimate intrinsic neural timescales (Fig. 1d)[2]. $R^2$ values were then used to quantify the goodness of fit between our model and the observed data (Fig. S1A). The exponential function was closely aligned with the autocorrelation profiles of each neuron (mean $R^2 = 0.78$; $R^2$ for Monkey $J = 0.73$; $R^2$ for Monkey $Q = 0.83$). Neurons with poor fits ($R^2 < 0.3$, $n = 20$) were excluded, leaving 380 neurons for analysis. For this final set of neurons, the precision of the fits was high with a median coefficient of variation (CV) of 0.22. Furthermore, a split-half analysis confirmed the stability of the measurements, showing strong positive correlations between the first and second halves of each recording session (Spearman's $\rho = 0.72$, $p < 2.2 \times 10^{-16}$; Fig. S2).

The distribution of intrinsic timescales among FEF neurons exhibited clear bimodality (Fig. 2a), confirmed by several statistical tests (Table S1). This bimodality was observed across both monkeys combined (Excess mass test: $p < 2.2 \times 10^{-16}$) and within each monkey individually (Monkey $Q$: $p < 2.2 \times 10^{-16}$; Monkey $J$: $p = 3 \times 10^{-2}$)[39]. Neurons were divided into short-$\tau$ and long-$\tau$ populations using the global minimum (60.95 ms) from the kernel density estimate of the combined distribution, which was consistent across individual monkeys (Monkey $J = 60.51$ ms; Monkey $Q = 59.33$ ms) (Fig. 2b). This division yielded a short-$\tau$ group ($N = 168$) with a mean timescale of $27.34 \pm 16.50$ ms and a long-$\tau$ group ($N = 212$) with a mean timescale of $102.64 \pm 23.31$ ms (Fig. 2c, d). Similar mean timescales were observed

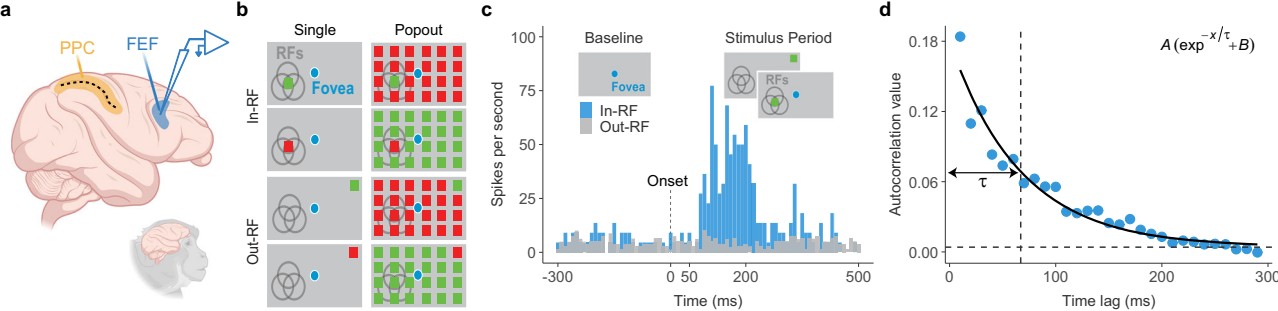

**Fig. 1 | Experimental setup and measurement of intrinsic neural timescales.** **a** Schematic of the experimental setup. Neuronal activity in the frontal eye field (FEF) was recorded with a linear electrode array under control conditions and during cryo-inactivation of the posterior parietal cortex (PPC). **b** Visual stimuli were either a single stimulus presented alone (Single) or the same stimulus embedded among uniformly colored distractors (Popout), appearing inside (In-RF) or outside (Out-RF) the neuron's receptive field. **c** Example neuronal response showing mean firing rate to a single stimulus inside (blue) or outside (gray) the receptive field

(10 ms bins). **d** Intrinsic neural timescales were estimated from spike-count autocorrelation during the baseline period. Blue dots indicate autocorrelation coefficients at successive 10 ms lags. Timescales were derived by fitting an exponential decay function (solid curve) to the autocorrelation data, where the decay constant reflects the intrinsic neural timescale ($\tau$; vertical dashed line). The horizontal dashed line indicates the offset of the exponential fit. Source data are provided as a Source Data file. **a** Created in BioRender. Soyuhos, O. (2026) https://BioRender.com/a85j026.

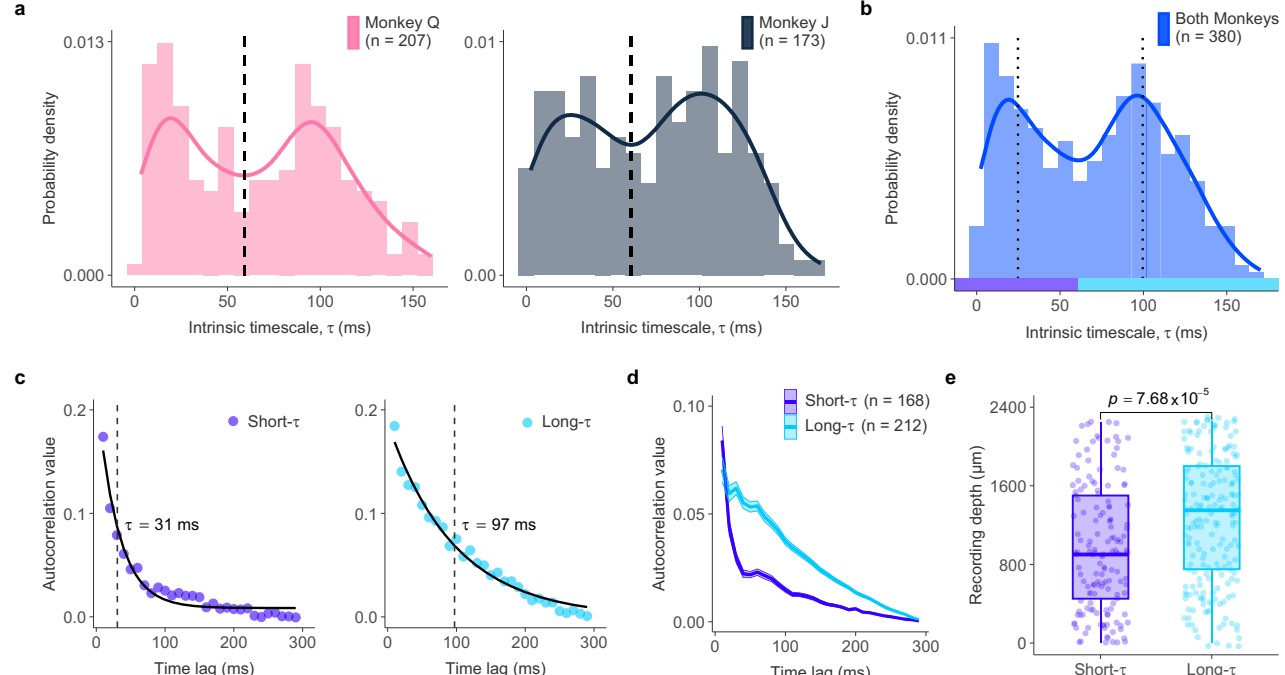

**Fig. 2 | Bimodal distribution of intrinsic neural timescales in FEF neurons.**
**a** Histograms and kernel density estimates of intrinsic neural timescales ($\tau$) for Monkey Q (left) and Monkey J (right). Solid lines show the kernel density estimates. Dashed lines mark the global minima separating short-timescale (short-$\tau$) and long-timescale (long-$\tau$) neurons. **b** Histogram and kernel density estimate of $\tau$ values for both monkeys combined. Dotted lines denote the median $\tau$ for each group.
**c** Example autocorrelation profiles from representative short-$\tau$ (left) and long-$\tau$ (right) neurons. Dots indicate autocorrelation values, the solid line shows the exponential fit, and the vertical dashed line marks the resulting intrinsic timescale

for each neuron. **d** Mean autocorrelation profiles for short-$\tau$ (purple) and long-$\tau$ (blue) neurons; shaded areas indicate ± SEM. **e** Comparison of recording depths for short-$\tau$ (purple) and long-$\tau$ (blue) neurons. Boxplots indicate the median (center line) and interquartile range (box; 25th–75th percentiles), with whiskers extending to 1.5 ×the interquartile range. Each point represents an FEF neuron (short-$\tau$: $n = 151$; long-$\tau$: $n = 199$). Group differences between short-$\tau$ and long-$\tau$ neurons were assessed with a two-sided Wilcoxon rank-sum test. Source data are provided as a Source Data file.

for both Monkey $J$ and $Q$ (short-$\tau$ neurons: Monkey $J$ = 28.17 ms; Monkey $Q$ = 26.71 ms; long-$\tau$ neurons: Monkey $J$ = 104.78 ms; Monkey $Q$ = 100.73 ms). In addition, we observed a significant depth-dependent separation between the two neuronal classes. Short-$\tau$ neurons were recorded at more superficial sites ($N = 151$; median depth = 901 $\mu$m), whereas long-$\tau$ neurons were found deeper ($N = 199$; median depth = 1351 $\mu$m; Wilcoxon rank-sum test, $p = 7.68 \times 10^{-5}$; Fig. 2e). The likelihood of encountering a long-$\tau$ neuron increased at deeper recording depths in the FEF (channels 8–16; 1050–2250 $\mu$m; Wilcoxon signed-rank test, $p = 3.59 \times 10^{-2}$; Fig. S3). Although absolute recording depth does not directly correspond to cortical layers due to the cortical fold and relative penetration angle, these results reveal a systematic depth-dependent organization of neurons with distinct intrinsic timescales.

This bimodal timescale distribution was evident within both the multi-unit ($N = 300$; Excess mass test: $p < 2.2 \times 10^{-16}$) and single-unit ($N = 80$; $p = 2 \times 10^{-3}$) populations (Fig. S4). Consistent with this finding, the distinction between timescale groups was not explained by unit type (Chi-squared test of independence: $\chi^2(1) = 1.38$, $p = 0.24$) or by differences in the precision of the timescale estimates (Wilcoxon rank-sum test: $p = 0.56$; median CV = 0.21 for short-$\tau$ vs. 0.24 for long-$\tau$).

In contrast to the bimodal distribution of timescales, firing rate distributions were consistently unimodal for both monkeys (Excess mass test: Monkey $Q$, $p = 0.43$; Monkey $J$, $p = 0.79$) (Fig. S5). However, the two groups differed in their intrinsic firing patterns. Short-$\tau$ neurons exhibited significantly higher temporal variability (Wilcoxon rank-sum test: $p = 4.6 \times 10^{-5}$; Fig. S6A) but similar trial-to-trial variability ($p = 0.31$; Fig. S6B) in their firing rate compared to long-$\tau$ neurons. This suggests that a short-$\tau$ signifies more transient activity characterized

by a dynamic baseline firing pattern, whereas a long-$\tau$ indicates sustained activity with a stable baseline firing pattern. Together, these results suggest the existence of at least two distinct neural circuit motifs within the FEF characterized by different temporal dynamics.

### Functional relevance of FEF intrinsic neural timescales
Next, we asked whether the intrinsic neural timescales measured during the task-free baseline period correlate with the neurons' functional properties during the task. We used single and popout indices to quantify neuronal visual responses and salience-driven modulation. Specifically, the single index was calculated by comparing the activity evoked by a single stimulus presented inside versus outside the receptive field (RF). This index captures the spatial selectivity of visual responses, reflecting the ability of neurons to distinguish stimuli within their RFs. Similarly, a popout index was calculated by comparing neuronal responses to unique stimuli inside versus outside the RF. This index quantifies the neural representation of stimulus salience, reflecting the ability of neurons to distinguish visually salient stimuli. To further characterize the temporal dynamics of these responses, we divided the neuronal activity into transient (50–200 ms) and sustained (200–500 ms) periods following stimulus onset. This temporal segmentation enabled us to calculate transient and sustained versions of each index, distinguishing short-lived neural responses to stimulus onset from long-term changes in activity associated with sustained stimulus processing (see Methods).

To isolate the relationship between intrinsic timescales and neuronal function, we used multiple linear regression to relate timescales to the single and popout indices from both transient and sustained epochs, while controlling for other predictors (Fig. 3a). For visual

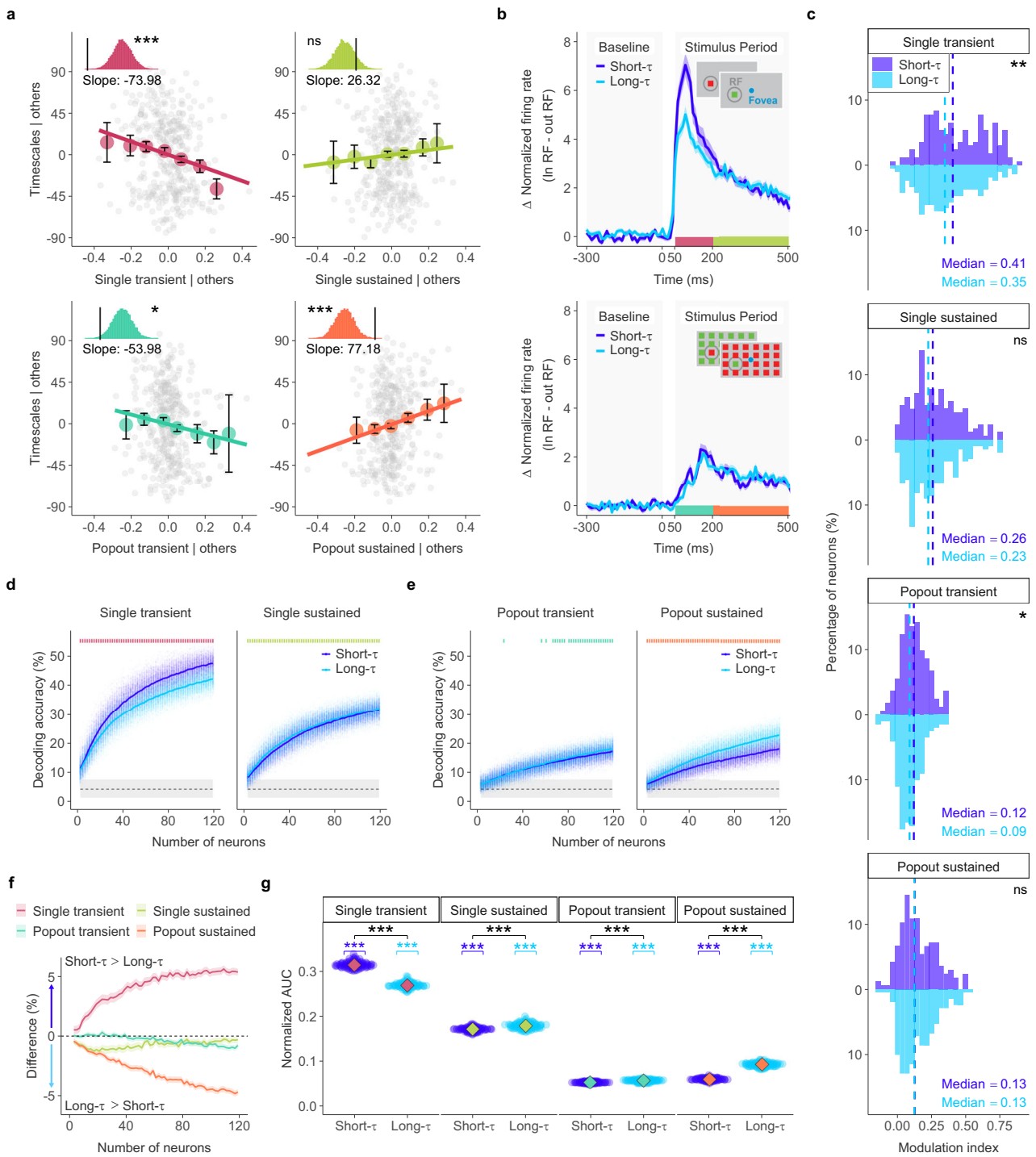

responses, we found a significant negative association between timescales and transient single indices (Permutation test: $p = 2 \times 10^{-4}$, $\beta = -73.98$). That is, neurons with faster intrinsic timescales showed stronger responses to single visual stimuli within their receptive fields than those with slower timescales. For salience processing, we found a significant negative relationship between timescales and transient popout indices (Permutation test: $p = 2.05 \times 10^{-2}$, $\beta = -53.98$), and a significant positive relationship with sustained popout indices ($p = 9 \times 10^{-4}$, $\beta = 77.18$). In other words, neurons with slower intrinsic timescales exhibited stronger sustained responses to popout stimuli, while neurons with faster timescales showed greater transient activity. These findings demonstrate a robust link between individual FEF

neurons' intrinsic timescales and their role in visual processing and salience representation.

Motivated by this functional relevance, we next compared visual responses and salience processing between short-$\tau$ and long-$\tau$ FEF neurons, defined by their intrinsic timescales during the task-free epoch (Fig. 3b). Short-$\tau$ neurons exhibited significantly greater transient single indices (Cramér–von Mises test: $p < 3 \times 10^{-3}$; median $= 0.41$ for short-$\tau$ vs. 0.35 for long-$\tau$) and transient popout indices ($p < 1.3 \times 10^{-2}$; median $= 0.12$ for short-$\tau$ vs. 0.09 for long-$\tau$) compared to long-$\tau$ neurons. In contrast, sustained indices were not significantly different for either single ($p = 0.16$) or popout ($p = 0.96$) conditions at the individual level (Fig. 3c). Additionally, the visual onsets for the two

**Fig. 3 | Intrinsic neural timescales correlate with the functional properties of FEF neurons. a** Added-variable plots showing the relationship between intrinsic timescales ($\tau$) and stimulus indices for single (top row) and popout (bottom row) conditions during transient (left) and sustained (right) periods. Gray points are individual FEF neurons ($n$ = 380); colored circles show binned mean values with error bars indicating mean ± SEM. Insets show the significance of the regression slope from two-sided permutation tests. **b** Differences in normalized responses to single (top) and popout (bottom) stimuli presented inside (In-RF) versus outside (Out-RF) receptive fields, shown separately for short-$\tau$ (purple) and long-$\tau$ (blue) neurons. Shaded areas indicate ± SEM. **c** Histograms of single and popout indices during transient and sustained periods, plotted separately for short-$\tau$ and long-$\tau$ neurons. Dashed lines mark median values. Differences between short-$\tau$ and long-$\tau$ distributions were assessed with Cramér–von Mises tests. **d** Population decoding accuracy for discriminating the location of a single stimulus out of 24 possibilities, plotted as a function of the number of neurons. Curves show mean accuracy for short-$\tau$ (purple) and long-$\tau$ (blue) populations during transient (left) and sustained

(right) periods. Surrounding points show accuracies from 500 decoding iterations. The bottom dotted line indicates chance performance (1/24); the gray shaded area shows its 95% CI. Vertical lines at the top mark neuron counts where the groups differ significantly, assessed with two-sided Wilcoxon rank-sum tests across iterations with $p$ values FDR-corrected for multiple comparisons. **e** Population decoding accuracy for discriminating the location of a popout stimulus, formatted as in (**d**). **f** Difference in decoding accuracy (short-$\tau$ minus long-$\tau$) between the two neuronal populations for each of the four conditions in (**d**, **e**). Shaded areas represent 95% CIs for the differences across iterations. **g** Comparison of normalized area under the curve (AUC) values from the decoding curves in (**d**, **e**) for short-$\tau$ and long-$\tau$ neurons. Normalized AUC values range from 0 (chance performance) to 1 (perfect decoding across all population sizes). Diamonds indicate medians. Within each condition, AUC values for short-$\tau$ and long-$\tau$ neurons were tested against zero using two-sided Wilcoxon signed-rank tests, and differences between groups were assessed with two-sided Wilcoxon rank-sum tests. Source data are provided as a Source Data file. (*, $p < 0.05$; **, $p < 0.01$; ***, $p < 0.001$; ns not significant).

groups did not significantly differ for either single (Cramér–von Mises test: $p = 0.2$) or popout stimuli ($p = 0.08$) (Fig. S7). These findings align with the regression analysis, demonstrating that neurons with shorter timescales exhibit stronger, but not faster, transient responses to both single and popout stimuli.

Next, we assessed the visual and salience information encoded at the population level by short-$\tau$ and long-$\tau$ neurons. We performed two complementary pseudo-population decoding analyses. The first was a binary stimulus detectability task, in which we predicted whether a stimulus location was inside or outside the receptive field. The second was a multi-class stimulus discriminability task, where we predicted the specific location of the stimulus out of 24 possibilities. We then generated neuron-dropping curves (NDCs) from neuronal subsets of each group to examine decoding accuracy across varying population sizes (Fig. S9 and Fig. 3d, e). To quantify the performance of each population, we calculated the area under these curves (AUC) that exceeded the chance level.

In the detectability task, we found a dissociation between the two populations (Fig. S9). For single-stimulus detection, short-$\tau$ neurons excelled during the transient period (Wilcoxon rank-sum test: $p = 2.41 \times 10^{-13}$, effect size = 0.60; median = 0.989 for short-$\tau$ vs. 0.984 for long-$\tau$), whereas long-$\tau$ neurons performed better during the sustained period ($p = 7.71 \times 10^{-3}$, effect size = 0.22; median = 0.953 for long-$\tau$ vs. 0.947 for short-$\tau$). For popout detection, long-$\tau$ neurons consistently showed stronger performance in both transient ($p = 3.94 \times 10^{-4}$, effect size = 0.29; median = 0.64 for long-$\tau$ vs. 0.62 for short-$\tau$) and sustained periods ($p = 1.88 \times 10^{-7}$, effect size = 0.43; median = 0.76 for long-$\tau$ vs. 0.73 for short-$\tau$).

This pattern of specialization was even more evident in the discriminability task (Fig. 3f, g). The single-stimulus discrimination results mirrored the detectability results: short-$\tau$ neurons again had better performance in the transient period ($p = 2.56 \times 10^{-34}$, effect size = 1.00; median = 0.31 for short-$\tau$ vs. 0.27 for long-$\tau$), while long-$\tau$ neurons were better during the sustained period ($p = 4.88 \times 10^{-15}$, effect size = 0.64; median = 0.18 for long-$\tau$ vs. 0.17 for short-$\tau$). This advantage for long-$\tau$ neurons was most apparent for popout discrimination, where they demonstrated better performance across both transient ($p = 1.44 \times 10^{-16}$, effect size = 0.68; median = 0.058 for long-$\tau$ vs. 0.054 for short-$\tau$) and sustained periods ($p = 2.56 \times 10^{-34}$, effect size = 1.00; median = 0.09 for long-$\tau$ vs. 0.06 for short-$\tau$). Overall, the differences between the groups were most pronounced for single-stimulus discrimination in the transient period and popout stimulus discrimination in the sustained period.

In summary, intrinsic neural timescales in the FEF are closely linked to visual responses and salience processing at both the individual and population levels, with short-$\tau$ neurons excelling in transient visual processing and long-$\tau$ neurons playing a greater role in sustained salience representation.

## Selective increase of FEF intrinsic neural timescales during PPC inactivation

We next investigated the effects of PPC inactivation on the FEF intrinsic timescales to understand how PPC regulates local neuronal dynamics within the FEF. PPC inactivation was achieved using cryoloops implanted in the IPS, which suppressed neuronal activity by lowering the temperature of the cortical tissue. The effectiveness of PPC inactivation was confirmed by robust behavioral effects in both the free viewing task and the double-target choice task, as reported by Chen et al.[19] Similar to the control condition, the exponential function showed a good fit to each neuron's autocorrelation profiles during inactivation (mean $R^2 = 0.8$; $R^2$ for Monkey $J = 0.75$; $R^2$ for Monkey $Q = 0.84$; Fig. S1B). The precision of the fits was also high (median CV = 0.24). Fig. 4a shows six representative neurons, illustrating the changes in their timescales during PPC inactivation. These neurons exhibited an increase in their intrinsic timescales, reflecting slower dynamics and a more stable firing pattern during the baseline period. Consistent with this, we found a significant increase in intrinsic timescales across the entire population of FEF neurons following PPC inactivation (Wilcoxon signed-rank test: $p = 5.09 \times 10^{-13}$, effect size = 0.43; inactivation: mean = 84.43 ms, median = 94.92 ms; control: mean = 69.89 ms, median = 73.14 ms) (Fig. 4b). This increase in intrinsic neural timescales was significant for both monkeys ($p = 3.02 \times 10^{-2}$, effect size = 0.19 for Monkey $J$; $p = 4 \times 10^{-15}$, effect size = 0.63 for Monkey $Q$) (Fig. S10). Thus, PPC inactivation robustly slowed neural dynamics across FEF neurons.

We further explored the changes in intrinsic timescales separately for the two distinct populations in FEF: short-$\tau$ and long-$\tau$ neurons. Similar to control conditions, we found that the probability density of FEF timescales exhibited a bimodal distribution during PPC inactivation (Excess mass test: $p < 2.2 \times 10^{-16}$). However, PPC inactivation led to a substantial increase (29.52%) in the number of long-$\tau$ neurons ($N_{Control} = 210$; $N_{Inactivation} = 272$) and a corresponding decrease (−38.27%) in the number of short-$\tau$ neurons ($N_{Control} = 162$; $N_{Inactivation} = 100$). This redistribution highlights a shift in the population dynamics toward slower timescales (Fig. 4c).

We then examined the magnitude of timescale changes separately for short-$\tau$ and long-$\tau$ neurons (Fig. 4d). We found that there was a significant increase in timescales for both short-$\tau$ neurons (Wilcoxon signed-rank test: $p = 1.02 \times 10^{-13}$, effect size = 0.67) and long-$\tau$ neurons ($p = 7.79 \times 10^{-3}$, effect size = 0.21). However, the changes observed in short-$\tau$ neurons were considerably larger than those of long-$\tau$ neurons. On average, short-$\tau$ neurons exhibited a 15-fold greater increase in timescales compared to long-$\tau$ neurons (Wilcoxon rank-sum test: $p = 8.64 \times 10^{-9}$, effect size = 0.35; short-$\tau$ neurons: $\Delta\tau$ mean = 30 ms, $\Delta\tau$ median = 19 ms; long-$\tau$ neurons: $\Delta\tau$ mean = 2 ms, $\Delta\tau$ median = 4 ms) (Fig. 4e). This effect was also observed individually in each monkey (Monkey $J$: $p = 3.65 \times 10^{-2}$, effect size = 0.19; Monkey $Q$: $p = 4.10 \times 10^{-9}$,

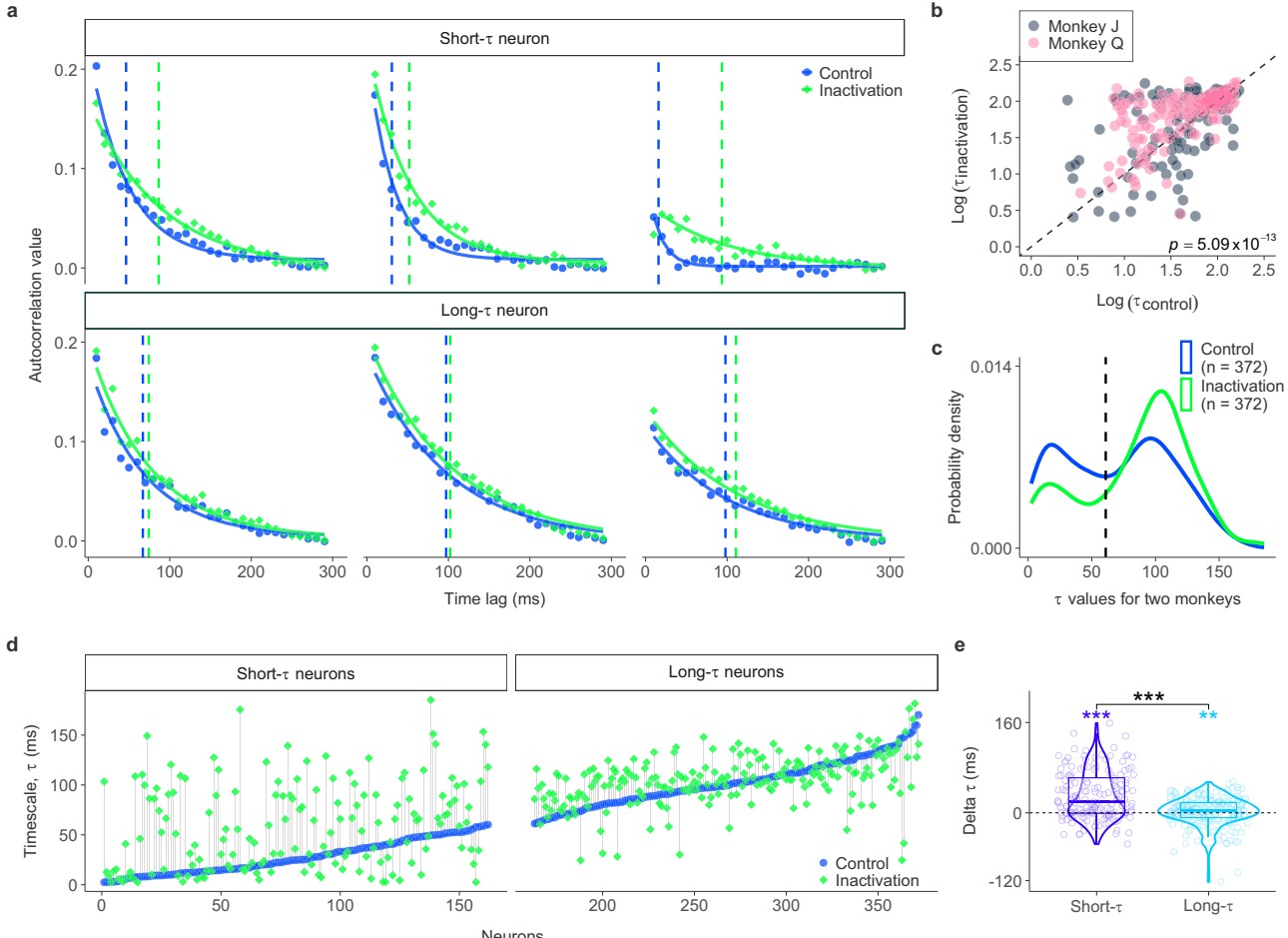

**Fig. 4 | PPC inactivation selectively alters intrinsic neural timescales of FEF neurons. a** Example autocorrelation profiles from representative short-τ (top) and long-τ (bottom) neurons under control (blue) and PPC inactivation (green) conditions. Dashed vertical lines indicate intrinsic timescales (τ). **b** Scatter plot comparing log-transformed τ values between control and PPC inactivation across both monkeys (Monkey *J*, gray; Monkey *Q*, pink). Differences between conditions were assessed with a two-sided Wilcoxon signed-rank test. **c** Probability density estimates of τ in control (blue) and PPC inactivation (green) conditions. **d** Individual neurons sorted by their τ under the control condition, shown separately for short-τ (left) and long-τ (right) neuronal groups. Blue and green markers indicate control

and PPC inactivation. **e** Violin plots showing changes in τ values (*Δτ* = inactivation − control) for short-τ (purple) and long-τ (blue) neurons. Violin shapes show the distribution of *Δτ*; overlaid boxplots indicate the median (center line) and interquartile range (box; 25th–75th percentiles), with whiskers extending to 1.5 × the interquartile range. Each point represents an FEF neuron (short-τ: *n* = 162; long-τ: *n* = 210). The dashed horizontal line marks zero change. Within each group, *Δτ* was tested against zero with two-sided Wilcoxon signed-rank tests, and differences between short-τ and long-τ *Δτ* distributions were assessed with two-sided Wilcoxon rank-sum tests. Source data are provided as a Source Data file. (**, *p* < 0.01; ***, *p* < 0.001).

effect size = 0.48) (Fig. S10C). Additionally, the changes in intrinsic neural timescales were not correlated with the changes in baseline firing rate (Spearman's rank correlation: $\rho = 0.25$, $p = 0.06$) (Fig. S11A). Although baseline firing rates slightly increased following PPC inactivation (Wilcoxon signed-rank test: $p = 5.94 \times 10^{-3}$, mean $\Delta$ firing rate = 0.93 Hz; Fig. S11B), these changes were similar between short-τ and long-τ neurons (Wilcoxon rank-sum test: $p = 0.33$) (Fig. S11C). These results indicate a selective dependence of FEF neural dynamics on PPC input.

## PPC inactivation selectively weakens the salience representation in FEF neurons

Finally, we assessed the effects of PPC inactivation on the relationship between intrinsic timescales and the functional properties of FEF neurons. We first re-examined the multiple linear regression analysis under PPC inactivation to determine if the input from PPC is necessary for the observed correlations between neural timescales and neuronal function (Fig. 5a). Our results revealed that the link between intrinsic timescales and salience representation is critically dependent on input from the PPC. Specifically, the significant negative association between

transient popout indices and timescales observed in the control condition (Permutation test: $p = 2.05 \times 10^{-2}$, $\beta = -53.98$) was no longer significant during PPC inactivation (Permutation test: $p = 0.63$, $\beta = -12.45$). Similarly, the significant positive association between sustained popout indices and timescales under control conditions (Permutation test: $p = 9 \times 10^{-4}$, $\beta = 77.18$) became nonsignificant during inactivation (Permutation test: $p = 0.21$, $\beta = -32.12$). For visual responses, PPC inactivation weakened but did not eliminate the significant negative correlation between timescales and transient single indices (Permutation test: $p = 2 \times 10^{-4}$, $\beta = -73.98$ under control; $p = 2.53 \times 10^{-2}$, $\beta = -44.83$ after inactivation). Thus, PPC inactivation exerted a larger impact on the correlation between neural timescales and salience representation compared to visual response.

We next assessed how PPC inactivation influenced visual responses and salience signals in individual short-τ and long-τ neurons by quantifying the change (*Δ*) in their single and popout indices. Consistent with the regression analysis, PPC inactivation exerted a significantly larger effect on popout indices than on single indices during the sustained period (Wilcoxon rank-sum test: $p = 2.17 \times 10^{-4}$, effect size = 0.22; median *Δpopout* = − 0.13 vs. median *Δsingle* = − 0.05). When comparing

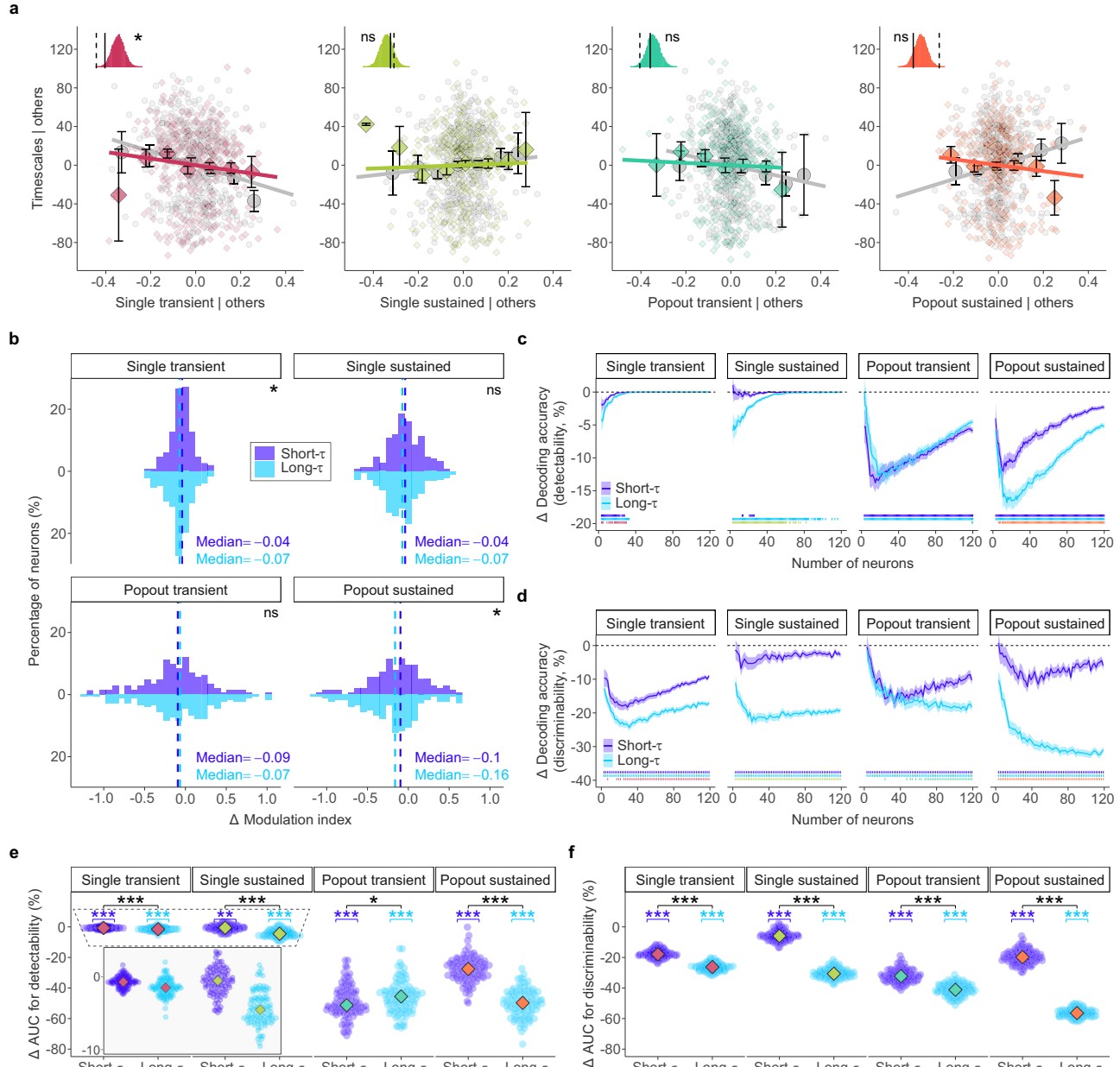

**Fig. 5 | Effects of PPC inactivation on visual responses and salience representation in FEF neurons. a** Added-variable plots showing relationships between intrinsic neural timescales (τ) and single/popout indices before (gray circles) and after posterior parietal cortex (PPC) inactivation (colored diamonds). Background points show individual FEF neurons in control (gray, *n* = 380) and PPC inactivation (color, *n* = 372); overlaid symbols show binned mean values with error bars indicating mean ± SEM. Insets depict slopes fitted to data before (solid lines) and after (dashed lines) PPC inactivation, compared with null distributions from two-sided permutation tests. **b** Change (Δ) in single and popout indices for short-τ (purple) and long-τ (blue) neuronal groups following PPC inactivation. Dashed horizontal lines indicate median Δ values. Differences between short-τ and long-τ distributions were assessed with Cramér–von Mises tests. **c** Neuron-dropping difference curves for the detectability task, showing relative change in decoding accuracy as a function of neuron number for short-τ (purple) and long-τ (blue) groups across the four stimulus conditions. Solid lines represent mean change and shaded areas

indicate 95% CIs. The horizontal dashed line at zero marks no change. Tick marks below each plot indicate population sizes with statistically significant changes: top row, short-τ; middle row, long-τ; bottom row, the difference between the two groups. These were assessed using two-sided Wilcoxon rank-sum tests across iterations with FDR correction for multiple comparisons. **d** Same as (**c**), but for the discriminability task. **e** Comparison of the relative change in normalized area under the curve (ΔAUC) for the detectability task between short-τ (purple) and long-τ (blue) neuronal groups. Each dot represents ΔAUC from a resampling iteration, and diamonds indicate medians. The inset magnifies changes in decoding performance for a single stimulus. Within each condition, ΔAUC values for each group were tested against zero using two-sided Wilcoxon signed-rank tests, and differences between groups were assessed with two-sided Wilcoxon rank-sum tests. **f** Same as (**e**), but for the discriminability task. Source data are provided as a Source Data file. (*, *p* < 0.05; **, *p* < 0.01; ***, *p* < 0.001; ns not significant).

between the neuron groups, this impairment was significantly greater in long-τ neurons for both transient single indices (Cramér–von Mises test: *p* = 1.50 × 10⁻²) and sustained popout indices (Cramér–von Mises test: *p* = 1.90 × 10⁻²), indicating that they are more susceptible to the loss of PPC input (Fig. 5b).

At the population level, we performed pseudo-population decoding analyses to quantify the change in decoding performance for short-τ and long-τ neurons (Fig. S12). For the detectability task, PPC inactivation impaired performance across all conditions (Fig. 5c). For single-stimulus detection, the impairment was significantly larger for long-τ neurons in

both the transient (−1.5% vs. −0.7% for short-$\tau$; Wilcoxon rank-sum test: $p = 4.77 \times 10^{-8}$, effect size = 0.45) and sustained periods (−4.3% vs. −0.7% for short-$\tau$; $p = 5.16 \times 10^{-21}$, effect size = 0.77). For popout stimulus detection, transient responses were slightly more impaired in short-$\tau$ neurons compared to long-$\tau$ neurons (−49.2% vs. −45.8%; $p = 2.4 \times 10^{-2}$, effect size = 0.18). However, during the sustained period, the impairment was substantially greater for long-$\tau$ neurons (−49.8% vs. −27.8% for short-$\tau$; $p = 4.2 \times 10^{-30}$, effect size = 0.93) (Fig. 5e).

This disproportionate impact on long-$\tau$ neurons was even more pronounced in the discriminability task (Fig. 5d). For single-stimulus discrimination, the performance reduction was significantly greater for long-$\tau$ neurons in both the transient (−26.4% vs. −17.7% for short-$\tau$; p = $3.63 \times 10^{-33}$, effect size = 0.98) and sustained periods (−30.3% vs. −5.8% for short-$\tau$; p = $2.56 \times 10^{-34}$, effect size = 1). This pattern was amplified for popout stimulus discrimination, where long-$\tau$ neurons again showed significantly larger deficits than short-$\tau$ neurons during both the transient (−41.2% vs. −32.1%; $p = 2.05 \times 10^{-27}$, effect size = 0.89) and, most notably, the sustained period (−56.3% vs. −19.6%; $p = 2.56 \times 10^{-34}$, effect size = 1) (Fig. 5f).

In summary, PPC inactivation disrupted the correlations between intrinsic neural timescales and salience representation in FEF. At both individual and population levels, removing PPC input predominantly impairs salience computation in long-$\tau$ neurons, with the most substantial deficits observed during the sustained period in both detectability and discrimination tasks.

## Discussion

We found that intrinsic neural timescales within the FEF exhibit a bimodal distribution, revealing two functionally distinct neuronal populations with fast (short-$\tau$) and slow (long-$\tau$) dynamics. These intrinsic timescales, measured during the baseline fixation period, correlated with neuronal functional properties. Specifically, short-$\tau$ neurons showed stronger transient responses to single visual stimuli, suggesting a role in rapid visual processing. In contrast, long-$\tau$ neurons exhibited stronger sustained responses to popout arrays, suggesting a role in sustained salience representation. Importantly, PPC inactivation selectively influenced these populations by predominantly increasing intrinsic timescales in short-$\tau$ neurons and substantially reducing salience signals in long-$\tau$ neurons. These findings demonstrate that PPC inputs selectively shape intrinsic neuronal dynamics and functional properties within the FEF, supporting distributed computations across the frontoparietal attention network.

Our results provide clear evidence for two functionally distinct neuronal classes in the FEF, defined by their intrinsic neural timescales. Notably, these classes differ primarily in their temporal dynamics, as their initial response latencies are comparable (Fig. S7). This finding aligns with recent reports of multimodal timescale distributions across the cortex in rodents, macaques, and humans[40,41]. Here, we demonstrate this dual-timescale organization in the FEF, a critical prefrontal oculomotor area in the frontoparietal attention network[15,18], and show that these two classes support a functional division of labor. This functional dissociation fits well within a theoretical core-periphery framework, which posits a trade-off between network stability and flexibility[42]. Short-$\tau$ neurons, with their shorter temporal correlation windows and higher temporal variability (Fig. S6), match the profile of a responsive "periphery" suited for rapidly updating visual information. In contrast, long-$\tau$ neurons, with their prolonged activity and lower variability, align with a stable "core" that supports the spatiotemporal integration required to maintain attentional priority[1,8]. Computationally, this heterogeneity likely underpins the functional flexibility required for the FEF's diverse cognitive roles, from visually guided saccades and spatial attention to decision-making and working memory[1,43–45].

It remains unclear how this heterogeneity in intrinsic timescales maps onto previously established functional and anatomical classifications of FEF neurons. Traditionally, FEF neurons have been categorized into visual, motor, visuomotor, and memory-delay neurons, based on their functional properties[43,46]. Visual neurons likely have shorter intrinsic timescales consistent with their strong transient visual responses, whereas memory-delay neurons presumably have longer timescales, aligning with previous studies[8,9,47]. Furthermore, these long-$\tau$ neurons may also play a dominant role in attentional deployment, modulating visual cortical activity via their direct axonal projections[19,48–50]. Our laminar analysis reveals a clear anatomical segregation that supports this functional division: short-$\tau$ neurons are located in more superficial layers, while long-$\tau$ neurons are predominantly found in deeper layers. This finding aligns with the general cortical motif where superficial layers are more strongly associated with feedforward processing and deeper layers with feedback and integration[51,52].

PPC inactivation had a selective impact on short-$\tau$ and long-$\tau$ neurons, suggesting at least two distinct network motifs within the FEF (Fig. 6). In this simplified circuit model, short-$\tau$ neurons exhibited strong transient visual responses but weak and short-lived salience representation, implicating their primary role in relaying quickly changing visual information (Fig. 6a). These neurons are likely feedforward recipients, receiving projections from the PPC and other posterior visual regions such as those in the occipital lobes[25,26], and have weaker recurrent connections with themselves and other PFC neurons. Consequently, PPC inactivation significantly altered their intrinsic dynamics but had a limited impact on their visual responses, possibly due to compensatory inputs from other visual regions. In contrast, long-$\tau$ neurons showed robust sustained salience representation when presented with popout arrays (Fig. 6b). The input from the PPC and reciprocal connections between the PPC and FEF might represent or resolve stimulus competition and play an important role in salience computation[14,19,53]. This functional role might be driven by their recurrent connectivity within PFC, which could promote sustained neural activity to maintain attentional modulation, including salience representations, over time[28,50]. Additionally, the longer intrinsic timescales characteristic of the PFC, driven by its recurrent connectivity, might dominate the dynamics of long-$\tau$ neurons in the FEF[2,8], thus mitigating PPC's influence on their intrinsic timescales. This would explain why PPC inactivation substantially impaired salience processing in these neurons, while having smaller effects on their intrinsic timescales. Future studies with large-scale recordings across both the PPC and FEF are essential to uncover the detailed circuit connections underlying these distinct neuronal populations.

Although we did not directly establish a causal link between timescale modulation and behavior, the observed neural changes may underlie the behavioral impairments seen in active tasks[19]. In the current study, we used a passive design without eye movement to isolate this underlying circuit mechanism, as active saccade tasks are known to introduce confounding motor-planning and motor-related signals that heavily modulate FEF activity[43,54]. In prior work using the same perturbation, PPC inactivation biased target selection and diminished the influence of visual salience on eye movements during free viewing of natural images: salience–fixation correlations were reduced for contralateral saccades (eye-centered) and for fixations landing in contralateral image regions (head-centered)[19]. The neurophysiological data reported here were collected in the same animals and during overlapping experimental sessions that included both the free-viewing and onset-synchrony paradigms. Together, these convergent neural and behavioral effects suggest a shared mechanism whereby PPC input regulates sustained salience representation within the FEF to support salience-driven behavior.

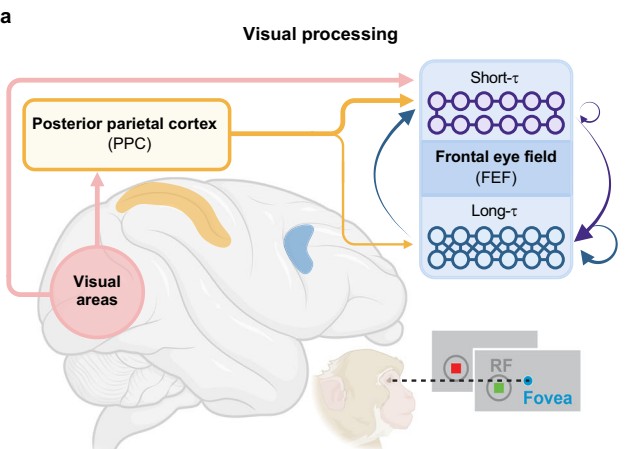

**Fig. 6 | Possible circuit model for visual processing and salience computation in the FEF. a** Short-timescale (short-τ) FEF neurons (blue) receive strong feedforward input from posterior parietal cortex (PPC; orange) and early visual cortex (pink). Because their local recurrence is comparatively weaker, their intrinsic dynamics are strongly shaped by PPC-dependent input (directly or via polysynaptic routes). These neurons primarily support transient, short-lived visual responses to single stimuli. **b** Long-timescale (long-τ) FEF neurons integrate information over extended temporal and spatial windows and show sustained salience-related modulation to popout arrays. Their activity is supported by stronger reciprocal PPC–FEF

interactions and robust recurrent prefrontal circuits, and they provide feedback signals to the visual cortex. Together, these properties position long-τ neurons to maintain attentional priority signals (such as visual salience) over time. Arrows indicate putative pathways; thickness reflects relative influence. The schematic summarizes functional relationships inferred from our data and prior work on the frontoparietal attention network and is not intended as a comprehensive anatomical map (for example, subcortical pathways are not depicted). Created in BioRender. Soyuhos, O. (2026) https://BioRender.com/g40r029.

Finally, our results provide direct evidence that long-range inter-areal inputs causally contribute to shaping intrinsic timescales in FEF. Because PPC is embedded within a distributed attention network, including subcortical areas[15,55], this regulation may arise via direct or polysynaptic pathways. However, anatomical studies indicate that the reciprocal projections between PPC (particularly LIP) and FEF are especially strong, comprising a major fraction of its long-range visual inputs[25,26]. Given this dominant connectivity, PPC inactivation is likely to exert a more substantial influence on FEF dynamics than perturbations of other individual inputs. Thus, our results provide a circuit-level mechanistic account of timescale control and align with computational frameworks in which intrinsic timescales emerge from the interaction of local recurrence and long-range connectivity[5,28,36–38]. We propose two complementary mechanisms for the observed slowing during PPC inactivation: (i) removal of relatively fast parietal input may unmask slower local dynamics in FEF, particularly in short-τ neurons with weaker recurrence; and (ii) inactivation may increase engagement of local recurrent circuits and broader prefrontal feedback loops, prolonging integration windows[38]. This framework yields testable predictions: removing inputs from areas with shorter intrinsic dynamics (e.g., parietal and extrastriate cortices) should lengthen FEF timescales, whereas perturbing long-timescale frontal inputs (e.g., supplementary eye field) should shorten FEF timescales by reducing recurrent drive. Projection-specific perturbations combined with simultaneous multi-area recordings will be crucial to quantify these relative contributions and to determine how intrinsic dynamics support attention and other cognitive functions.

## Methods

### Subject details

All experimental procedures followed the guidelines set by the National Institutes of Health Guide for the Care and Use of Laboratory Animals, the Society for Neuroscience Guidelines and Policies, and the Stanford University Animal Care and Use Committee. The experiments involved two healthy adult male rhesus macaques (*Macaca mulatta*) obtained from the California National Primate Research Center (UC Davis): monkey *Q* (16 kg) and monkey *J* (17 kg). The number of animals

used aligns with typical practices for neurophysiological studies involving non-human primates.

### Experimental procedures

The details of the surgical procedures and neurophysiological recordings have been previously reported by Chen et al.[19]. In summary, two monkeys (*Q*, 16 kg; *J*, 17 kg) performed visual sensitivity and salience-driven attention tasks under PPC inactivation and control conditions while we monitored neural activity in the FEF (Fig. 1a). Eye position was monitored with an EyeLink 1000 Plus eye tracker (SR Research). Each trial began with an initial central fixation period (1 × 1 dva) of 500 ms on a gray background (60 cd/m²), with the last 300 ms serving as the baseline period. Following this, the monkeys were presented with either a single visual stimulus or a popout stimulus for 500 ms while continuing to fixate at the center (Fig. 1b). In the single-stimulus trials, a red or green square stimulus (7 × 7 dva) was displayed at one of 24 locations on a 6 × 4 grid (75 × 45 dva). In the popout trials, a square stimulus was placed among an array of stimuli with contrasting colors (e.g., a red square among green stimuli or a green square among red stimuli). After stimulus offset, the monkeys continued to fixate for an additional 500 ms to receive a reward (Fig. 1c). On average, there were 20 trials per location for each stimulus type in both the control and inactivation conditions.

### Neurophysiological recording of FEF neurons

The recording locations within the right FEF were identified by eliciting short-latency saccadic eye movements using biphasic current pulses (≤50 µA; 250 Hz; 0.25 ms duration), in line with methods from previous studies[56]. Neural activity was recorded using 16- or 32-channel linear array electrodes (V and S-Probes, Plexon, Inc.), with contacts spaced 150 µm apart. These electrodes were inserted into the cortex via a hydraulic microdrive (Narishige International). Neural signals were referenced to a nearby stainless steel guide tube positioned near the electrode contacts and acquired using an OmniPlex Neural Data Acquisition System (Plexon, Inc.). The signal used for spike detection was filtered with a 4-pole Bessel high-pass filter at 300 Hz and a 2-pole Bessel low-pass filter at 6000 Hz, and sampled at 40 kHz. Initial

classification of extracellular waveforms into single neurons or multi-units was performed online via template matching, with offline sorting subsequently verified using Offline Sorter (v3.0; Plexon, Inc.). In total, we recorded spike timestamps from 400 neural units in the FEF across the two monkeys.

## Reversible inactivation of PPC

In each monkey, a pair of stainless steel cryoloops was implanted in the intraparietal sulcus (IPS) of the right hemisphere. These cryoloops were custom-designed to fit the contours of the IPS and fill the sulcus. The IPS was cooled by circulating chilled methanol through the loop tubing, with the temperature closely monitored and maintained within 1 °C of the target value by adjusting the methanol flow rate. A stable loop temperature of approximately 5 °C was reached within 5–10 min after the cooling process began, and the brain's normal temperature was restored within about 2 min post-cooling. Temperatures around 5 °C in the loop have been shown to effectively suppress neuronal activity in the underlying cortex[57]. During the experimental sessions, inactivation periods lasted 30–60 min each. Neurophysiological data were first collected during a control block, followed by an inactivation block across eight sessions. In six additional sessions, data collection followed two consecutive sets of control and inactivation phases (control–inactivation–control–inactivation), allowing the examination of potential block order effects. The success of PPC inactivation on behavior was evaluated in separate sessions using double-target choice tasks and free viewing of natural images[19].

## Computation of intrinsic neural timescales

We analyzed neural activity from 400 neural units in the FEF across two monkeys. Each unit's activity was recorded as spike timestamps, which were binned into 10 ms intervals to calculate spike counts for each trial. To normalize the spike counts at each time bin across all trials, we subtracted the mean firing rate of each time bin from the spike count in that bin. This step helped isolate the intrinsic temporal dependencies of the neuron's firing pattern, independent of its overall firing rate. We then performed autocorrelation analysis on the 300 ms baseline windows of each trial to compute the sum of products of spike count values against themselves at various lags for each neuron. The autocorrelation values for each lag were first averaged across trials for each neuron. Next, these averaged autocorrelation values were scaled by dividing them by the zero-lag value, standardizing the results for each neuron.

We then fitted a single exponential decay function to these averaged autocorrelation values and calculated the decay time constants for each neuron, representing their intrinsic neural timescales (Fig. 1d). Importantly, a subset of neurons displayed significantly low autocorrelation values at short time lags, suggesting the presence of refractory periods or negative adaptation, as reported in previous studies[2,6,8]. Therefore, the fitting procedure was started at the time lag where the mean autocorrelation reached its maximum value (excluding the zeroth lag where autocorrelation equals one). The initial dip in autocorrelation profiles did not systematically differ between short-$\tau$ and long-$\tau$ groups; neither the magnitude at the first time lag ($p = 0.31$) nor at the subsequent peak ($p = 0.17$) differed significantly (Wilcoxon rank-sum test).

For the curve fitting, we used the Trust Region Reflective algorithm. The model used for the fitting process is given by the equation:

$$f(x) = A \cdot \left( e^{-x/\tau} + B \right) \tag{1}$$

where $x$ is the time lag, $\tau$ represents the intrinsic timescale, $A$ is the amplitude of the decay, and $B$ is included to adjust for baseline shifts, addressing the influence of intrinsic neural timescales that extend beyond the measurement window.

To establish an objective criterion for the goodness of fit, we calculated pseudo-$R^2$ values for nonlinear fitting based on the proportion of the total variance explained by the exponential decay function. We found that 5% of units had $R^2$ values below 0.3, indicating a poor fit. Additionally, 6.5% of units fell within the range of 0.3 to 0.5, representing a moderate fit, and 13% of units had values between 0.5 and 0.7, suggesting a good fit. The majority, 75.5% of units, demonstrated values ranging from 0.7 to 1.0, indicating an excellent model fit (Fig. S1). Neurons with $R^2$ values below 0.3 were excluded from further analysis, resulting in a remaining sample of 380 neural units for the control condition and 372 neural units for the inactivation condition. In addition, the uncertainty of each timescale estimate was quantified by the coefficient of variation (CV; standard error / $\tau$), derived from the variance of the exponential fit. The resulting median CV of 0.22 confirmed the high precision of the timescale estimates.

Finally, we tested the stability of our timescale measurements using a split-half analysis. For each neuron, trials within a recording session were divided into first and second halves, and an intrinsic timescale was calculated for each subset independently using the procedure described above. We then used Spearman's rank correlation to quantify the consistency of timescales between the two halves (Fig. S2).

## Mode testing of intrinsic neural timescales

To investigate the distribution of intrinsic neural timescales in the FEF, we applied several statistical tests using the `mode` test function from the `multimode` R package[58]. Each test was chosen for its ability to highlight different aspects of distribution modality. We used the Hall and York Critical Bandwidth Test[59], which refines bandwidth selection in kernel density estimation to improve the detection of subtle multimodal patterns. The Fisher and Marron Cramér–von Mises Test[60] was applied for its sensitivity in capturing subtle features like shoulder modes by assessing how well the empirical distribution function fits a hypothesized unimodal function using the Cramér–von Mises statistic. Additionally, we used the Excess mass test[39], which employs the excess mass criterion for mode detection, offering a robust method particularly suited for multidimensional datasets with high noise levels. All tests provided complementary results (Table S1).

## Analysis of laminar position

We analyzed the laminar depth of each recorded neuron to infer whether the short-$\tau$ and long-$\tau$ classes were located at different cortical depths. This analysis was based on data from 12 recording sessions. In 11 of these sessions, we used 16-channel linear electrode arrays, while one session utilized a 32-channel array. The 16-channel configuration (150 $\mu$m spacing) spans 2.25 mm, a depth that corresponds well with the known thickness of the macaque FEF (approximately 2.0–2.5 mm). To ensure our measurement window was consistent and comparable across all 12 sessions, we restricted our analysis to neurons recorded from the first 16 channels, with the uppermost channels consistently positioned near the cortical surface. This resulted in the exclusion of 30 neurons recorded on deeper channels from the single 32-channel session. The depth of each neuron was then compared between the short-$\tau$ and long-$\tau$ groups using a nonparametric Wilcoxon rank-sum test.

## Calculation of single and popout indices

We measured the receptive field of each neuron by analyzing their responses to a single stimulus presented at each of 24 locations. Locations where a neuron's firing rate exceeded 95% of its maximum response were included within the neuron's receptive field. In contrast, the four locations in the far-right column of the ipsilateral visual field were considered outside the receptive field. Following this, we computed visual indices for each neuron, reflecting the neural activity

during transient (50–200 ms) and sustained (200–500 ms) periods for both single and popout stimulus trials. These indices were calculated based on the average differential spike rate when stimuli were presented within versus outside the neuron's receptive field (In-RF vs. Out-RF; Fig. 1b), normalized by their sum:

$$\text{Visual index}_{i, \text{type}, \text{period}} = \frac{\overline{\text{In-RF}}_{i, \text{type}, \text{period}} - \overline{\text{Out-RF}}_{i, \text{type}, \text{period}}}{\overline{\text{In-RF}}_{i, \text{type}, \text{period}} + \overline{\text{Out-RF}}_{i, \text{type}, \text{period}}} \quad (2)$$

Here, $i$ represents each neuron, "type" refers to single or popout trials, and "period" denotes the transient or sustained epoch. We computed four single and popout indices corresponding to transient and sustained visual responses as well as salience representation. The distributions of single and popout indices in short-$\tau$ and long-$\tau$ neurons were compared using the Cramér–von Mises test (Fig. 3c).

## Analyzing visual onsets

We quantified the earliest time at which each neuron's activity rose significantly above baseline (onset latency), separately for single and popout stimulus conditions. Specifically, we computed the mean spike count across trials for each 10 ms bin. A baseline threshold was then defined by adding two standard deviations to the average spike count during the pre-stimulus window. Within the post-stimulus period, we then identified the first occurrence of at least two consecutive bins (20 ms) exceeding this threshold. The leading bin in this sequence was determined as the neuron's onset bin. By applying this procedure independently to single and popout stimulus trials, we observed that 154 of 168 short-$\tau$ neurons (92%) and 173 of 212 long-$\tau$ neurons (82%) showed a detectable onset under single-stimulus trials. Additionally, 145 of 168 short-$\tau$ neurons (86%) and 142 of 212 long-$\tau$ neurons (67%) showed a detectable onset during popout trials. These results provided onset latencies that characterized the timing of visually evoked responses across stimulus and experimental conditions (Fig. S7). Visual inspection of the neuronal activity traces confirmed the accuracy of these onset calculations.

## Multiple linear regression analyses

We conducted multiple regression analyses to assess the individual contributions of each variable while accounting for their interdependencies. In our model, the dependent variable was the intrinsic neural timescale, while the independent variables included the single indices, popout indices, and the baseline mean firing rate. The regression model was specified as:

$$\begin{aligned}
\tau_i = \beta_0 &+ \beta_1 \cdot \text{Single-transient}_i + \beta_2 \cdot \text{Single-sustained}_i \\
&+ \beta_3 \cdot \text{Popout-transient}_i + \beta_4 \cdot \text{Popout-sustained}_i \quad (3) \\
&+ \beta_5 \cdot \text{Firing-rate}_i + \epsilon_i
\end{aligned}$$

In this equation, $\tau_i$ is the intrinsic timescale for neuron $i$. The predictors are the transient and sustained single (Single-transient$_i$, Single-sustained$_i$) and popout (Popout-transient$_i$, Popout-sustained$_i$) indices, along with the baseline mean firing rate (Firing-rate$_i$). The term $\epsilon_i$ represents the residual error for each neuron.

To fit the linear model, we used robust regression, which minimizes the influence of outliers and is less sensitive to non-normality in the data. This approach ensures that the coefficients accurately reflect the effect of each predictor on the dependent variable without being disproportionately influenced by outliers. We utilized added-variable plots to illustrate the relationship between neural timescales and each predictor by plotting the controlled residuals against each independent variable (Fig. 3a). These plots provide a visual representation of the distinct contributions of each predictor to the model, with the slopes quantifying the effect of a one-unit change in each predictor on the timescales while holding the other variables constant.

We checked for multicollinearity among the independent variables in the multiple regression model using the variance inflation factor (VIF) analysis. The VIF quantifies the degree to which the variance of a regression coefficient is inflated due to linear dependence on other predictors in the model. Generally, a VIF value greater than 5 or 10 suggests problematic multicollinearity[61], indicating that the affected predictors may be linearly dependent on others. In our analysis, we did not find significant multicollinearity (Fig. S8A).

Additionally, we examined the normality of the residuals in our model. A Q–Q plot was used to visually assess the distribution of residuals from our robust regression model against a theoretical normal distribution (Fig. S8B). Deviations from the line in this plot indicate departures from normality, suggesting that the residuals were not normally distributed. To address this, we used permutation testing to assess the significance of the regression coefficients. We employed a custom function to randomly shuffle the dependent variable (neural timescales) 10,000 times while keeping the independent variables unchanged. This generated a distribution of regression coefficients under the null hypothesis of no association (Fig. 3a). $P$ values were calculated as the proportion of permuted coefficients whose absolute values equaled or exceeded that of the observed coefficient.

## Neuron-dropping decoding analyses

To assess whether short-$\tau$ and long-$\tau$ neurons differ in visual responses and salience representation at the population level, we conducted a neuron-dropping analysis[62–64] (Fig. 3d, e). The general framework was applied to two distinct decoding tasks: a binary detectability task and a multi-class discriminability task. For both, we evaluated decoding during transient and sustained stimulus presentations in single-stimulus and popout trials. The core methodology involved analyzing subpopulations of increasing size from 3 to 120 neurons; subpopulation sizes were incremented by one neuron for the detectability task and by two neurons for the discriminability task. The cap of 120 neurons was set to avoid statistical issues that can arise when the subpopulation size approaches that of the fixed population[62]. At each step, we randomly sampled the specified number of neurons 500 times to ensure robust performance estimates. For each sample, neural data underwent a principal component analysis (PCA) where the top three components were retained, and these components were then used to train a support vector machine (SVM) classifier.

The two decoding tasks differed in their specific objective and trial balancing. For both tasks, neurons with fewer than 20 trials for any condition were excluded. The detectability task tested the ability to decode spatial location as either inside vs. outside the receptive field. For this task, trial counts were equalized for In-RF and Out-RF conditions. This analysis included 143 neurons in the short-$\tau$ group and 190 neurons in the long-$\tau$ group. The discriminability task, in contrast, tested the ability to decode the precise location of the stimulus from 24 possible locations. For this multi-class problem, trial counts were equalized across all stimulus locations for each subpopulation. This analysis resulted in a final pool of 141 short-$\tau$ and 178 long-$\tau$ neurons. For both tasks, decoding performances exhibited saturation curves as the number of neurons increased, indicating diminishing returns in performance improvement with additional neurons.

To assess differences in decoding performance between short-$\tau$ and long-$\tau$ neurons, we analyzed the area under the curve (AUC) of these saturation curves. First, a statistical baseline was established by computing the 95% confidence interval (CI) of chance performance for each neuron count. This CI was calculated from the 2.5th and 97.5th percentiles of a binomial distribution, with the number of trials at each population size estimated by simulating the neuron and trial sampling procedure 1000 times. For the detectability task, the chance probability was $p = 0.5$, whereas for the discriminability task, it was $p = 1/24$, where 24 is the number of stimulus locations. The AUC was then calculated as the area between the decoding accuracy curve and the

upper bound of this chance-level CI, using the trapezoidal rule. To generate a distribution of AUC scores, the 500 resampling iterations were randomly partitioned into 100 non-overlapping groups, and the AUC was computed for the average decoding curve of each group. These raw AUC values were then normalized by dividing each by the maximum possible area above the chance CI (i.e., the area between a perfect 100% accuracy curve and the CI's upper bound). This yields a normalized score from 0 to 1, where 0 represents performance indistinguishable from chance and 1 represents perfect decoding across all population sizes. The resulting normalized AUC scores were compared between short-$\tau$ and long-$\tau$ neurons using the nonparametric Wilcoxon rank-sum test.

### Testing the effect of parietal inactivation

To investigate the effects of parietal inactivation on intrinsic neural timescales, visual responses, and attentional modulation in the FEF, we conducted a series of statistical analyses. First, we performed paired Wilcoxon signed-rank tests to determine whether parietal inactivation led to significant changes in FEF neural timescales, assessing the direction of change (increase or decrease) across both monkeys and within each individual monkey. The rank biserial correlation effect size was calculated to quantify the magnitude of the observed changes (Fig. 4b). Next, we analyzed whether the change ($\Delta$: inactivation−control) in neural timescales differed between short-$\tau$ and long-$\tau$ neurons. We first tested against zero using a one-sample Wilcoxon signed-rank test to determine whether the change within each group was significant, followed by unpaired Wilcoxon rank-sum tests to compare the magnitude of change between the two groups (Fig. 4e).

We also examined whether the relationships between neural timescales and visual/attentional responses observed in the control condition persisted or were altered during inactivation. A multiple regression analysis was conducted using values obtained after inactivation, with intrinsic neural timescales as the dependent variable and visual indices along with baseline firing rate as independent variables (Fig. 5a). Additionally, changes in the distribution of single and popout indices were compared between short-$\tau$ and long-$\tau$ neurons using the Cramér−von Mises test (Fig. 5b). Furthermore, we performed a neuron-dropping analysis[62] under PPC inactivation to evaluate the changes in decoding performances at the population level (Fig. 5c, d and Fig. S12). We compared the relative change in the normalized AUC scores between the two groups following PPC inactivation (Fig. 5e, f).

### Reporting summary

Further information on research design is available in the Nature Portfolio Reporting Summary linked to this article.

## Data availability

The data generated in this study have been deposited in the Figshare database (https://doi.org/10.6084/m9.figshare.28971866). Source data are provided with this paper.

## Code availability

The analysis code used to perform all statistical analyses and generate all figures is publicly available via a Code Ocean capsule (https://codeocean.com/capsule/0814945). The capsule includes the code and the dataset needed to reproduce the analyses and figures (R v4.4.1; RStudio v2024.4.2.764).

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

## Acknowledgements

We thank Dr. Tirin Moore for his contributions to the initial experimental design, securing funding for the project, and for providing constructive feedback. This work was supported by the National Science Foundation under grants 2152260 (NSF NRT NeuralStorm) (to O.S.) and 2207895 (to R.C.), National Institutes of Health grants EY014924 (to X.C.) and EY029759 (to T.M.), the Brain and Behavior Research Foundation (to X.C.), the Alfred P. Sloan Foundation under Grant No. FG-2021-16304 (to R.C.), and the UC Davis Large Interdisciplinary Applications in Neuroscience (LIAN) (to R.C. and X.C.).

## Author contributions

Conceptualization: X.C., O.S., and R.C.; Experimental design: X.C., M.Z., and T.M.; Performing experiments: X.C. and M.Z.; Methodology: O.S., X.C., and R.C.; Software: O.S.; Formal analysis: O.S.; Visualization: O.S.; Writing – original draft: O.S. and X.C.; Writing – review & editing: O.S., X.C., R.C., and T.M.; Funding acquisition: X.C., T.M., O.S., and R.C.

## Competing interests

The authors declare no competing interests.
