## [Transparent Peer Review file · Nature Communications]

Selective control of prefrontal neural timescales by parietal cortex

Corresponding Author: Dr Xiaomo Chen

Version 0:

Reviewer comments:

Reviewer #1

(Remarks to the Author)

In this manuscript, Soyuhos et al. have investigated intrinsic timescales in the frontal eye field (FEF), an area that plays a key role in stimulus driven and goal directed visual attention. They identified two distinct neuronal populations within FEF — one with short and another one with long intrinsic timescales — that contribute differentially to transient stimulus processing and attentional functions, respectively. To test whether these timescales are influenced by factors beyond local circuitry, the authors inactivated the posterior parietal cortex (PPC) using cryoloops implanted in the intraparietal sulcus (IPS). This manipulation significantly reduced the number of short-timescale neurons and increased the number of long-timescale neurons in FEF, resulting in an overall shift toward longer intrinsic timescales. Additionally, PPC inactivation selectively disrupted the attentional modulation in long τ neurons. These findings highlight the importance of interareal interactions, demonstrating that intrinsic timescales are not solely shaped by local network interactions but also by input from other brain regions.

While the study provides important insights on the heterogeneity of time intrinsic time scales with FEF, it lacks important large-scale population and laminar analysis. Such data and analysis could potentially offer a more comprehensive understanding of the spatial distribution and population-level dynamics/computations associated with the distinct neuronal classes identified in this study.

Regarding novelty and significance, previous studies have already reported the heterogeneity in time scales within cortical areas, and those studies are cited. They have now shown that this also holds for FEF, an area important for visual attention. Their second finding, the influence of PPC on intrinsic timescales in FEF, is the real new finding. However, one cannot rule out the effect of other brain areas that might have contributed to the change in time scales with the inactivation of PPC. Hence the changes they report might not be exclusively determined by PPC.

Major comments:

1. Do the two distinct neuron types based on intrinsic timescales form spatially segregated populations, and is there higher/lower functional coupling within/between these populations? Additionally, I am wondering whether there is a laminar dependence of the location of these distinct classes of neurons which could be important given how the input from PPC modulates their dynamics. Given that the authors recorded using laminar probes, in principle it should be possible to extract that information from the data.
2. In the decoding analysis, the decoder performance is already highly saturated for the classes used (in/out of RF). Wouldn't it be more appropriate to use a set of stimuli try to decode stimulus information from neural responses of these 2 groups of neurons?
3. The authors should include representative autocorrelograms for the two neuronal classes in Figure 1 as it will enhance the reader's understanding of intrinsic timescales by providing visual intuition
4. The authors mention that the FEF receives inputs from multiple brain areas. In this study they inactivated one of its inputs from the PPC. Do the authors have any mechanistic understanding on how inactivation of other brain areas might influence intrinsic timescales, and how these effects might compare in magnitude to the PPC inactivation results?

5. Is there a possibility that inactivating PPC might affect other brain areas which in turn feed into FEF? As a result, can the findings reported here be the combined effect of indirect perturbation of other areas in addition to PPC? These possibilities are not adequately considered, which weaken the authors' conclusions.

(Remarks on code availability)

Reviewer #2

(Remarks to the Author)

In this study, Syuhos et al. investigate how intrinsic neural timescales vary across neurons in the frontal eye field (FEF), considering both stimulus-driven and attention-modulated responses, and how these timescales are affected by inactivation of the posterior parietal cortex (PPC)—a region known to regulate attentional signals in FEF. Intrinsic timescales were estimated via the autocorrelation decay of spike-count fluctuations during spontaneous activity (i.e., the baseline period preceding stimulus onset).

The authors identified two distinct classes of FEF neurons based on their intrinsic timescales: short (<50 ms) and long (50–150 ms). When analyzing the neurons' responses to visual stimuli, they found that short-timescale neurons were more active during the transient, early phase of the response, whereas long-timescale neurons exhibited stronger attentional modulation during the later, sustained phase. PPC inactivation led to an overall increase in FEF timescales, with a stronger effect observed in short-timescale neurons. Notably, attentional modulation was reduced following PPC inactivation, especially in long-timescale neurons.

I found this study compelling. It is clearly written, the methods are well described, and the results are appropriately analyzed and interpreted. Nonetheless, I have a few concerns regarding the stability of the intrinsic timescale estimates and the theoretical framing of the findings:

1. Stability of intrinsic timescales: The robustness of the estimated timescales should be assessed. For instance, the authors could divide each recording session into two halves and compute timescales separately to test their consistency (e.g., via a split-half correlation). This would help ensure that the timescale classification reflects stable intrinsic properties rather than session-specific noise.

2. Theoretical framework: The study would benefit from a broader theoretical context. Recent work (Ponce-Alvarez et al., 2025, DOI: 10.1523/JNEUROSCI.1699-24.2024) has shown that neural timescales and variances relate systematically to a network's topology and functional roles. Specifically, neurons at the core of the network exhibit longer timescales, lower variability, and weaker stimulus responses, while peripheral neurons show shorter timescales and greater responsiveness. This reflects a fundamental trade-off between stability and flexibility in network dynamics. Framing the current results within this context could strengthen the interpretation of the timescale classes and their functional relevance.

3. Additional analyses: Following Ponce-Alvarez et al. (2025), it would be informative to analyze the intrinsic variance and pairwise correlations among neurons (and how they change during PPC inactivation). One might expect short-timescale neurons to exhibit higher variability and weaker correlations, whereas long-timescale neurons may show lower variance and stronger correlations among them. Exploring these dimensions could provide a more comprehensive characterization of the functional organization of the FEF.

Other comments:

- It seems that some neurons displayed significantly low autocorrelation values at short time lags, which led the authors to fit exponential functions to the autocorrelation functions (ACFs) for non-zero lags. Are these particular neurons classified as belonging to short- or long-timescale group?

- The uncertainty of the estimated ACF decay (τ) should be reported. Additionally, it would be informative to examine how this uncertainty varies as a function of the timescale.

- Figures 5C and 5D are difficult to interpret and could be improved for clarity. In particular, the use of a secondary axis should be avoided, as it is confusing.

- I could not find the number of trials for each condition.

- For the decoding, take into account the confidence interval of the chance level (which depends on the number of trials).

I hope the authors find these comments constructive.

(Remarks on code availability)

The legibility of the provided code could be improved, although it appears to contain all the necessary information to reproduce the results. A README file is included with sufficient instructions for installing and running the application. I did not attempt to install or run the code.

Reviewer #3

(Remarks to the Author)

The authors demonstrate that within the same cortical area, the frontal eye field, there are populations of neurons operating

at a shorter and a longer time scale; these populations have distinct functional properties, and are differentially affected by inactivation of the posterior parietal cortex.

Experiments are clearly designed; results are solid and clearly presented.

I have only minor comments/suggestions.

Species might be indicated in the title or abstract

Fig 1 legend

or the same stimulus embedded among iden- 657
tically colored distractors ("Popout")

-> distractors of a different uniform color (or other wording? now, "identically colored" may refer to either stimulus or distractors).

Fig. 1 C,D - time bin 10 ms?

Fig 6. Legend:

Short-timescale (short- τ) neurons in the frontal eye field (FEF; blue) receive direct feedfor- 736
ward inputs from the posterior parietal cortex (PPC; orange) and early visual cortical areas (pink). 737

in the scheme, there is no direct Visual -> FEF connection; only via PPC.

(Remarks on code availability)

code will be made publicly available after acceptance of the ms

Reviewer #4

(Remarks to the Author)

In this study, the authors present a novel analysis of a dataset originally collected by Chen et al. (Neuron, 2020), in which two monkeys performed visual sensitivity and saliency-driven attention tasks under both PPC-inactivation and control conditions. Neural activity was recorded in the frontal eye fields (FEF), revealing that neurons with shorter intrinsic timescales exhibited stronger transient responses to visual stimuli, suggesting a role in rapid visual processing. In contrast, neurons with longer intrinsic timescales showed more sustained responses, indicating their involvement in maintaining attention over time.

Importantly, PPC inactivation caused a lengthening of the shorter intrinsic timescales and reduced attentional modulation in neurons with longer timescales, considering both multi-unit and single-unit activity (namely, MUA and SUA respectively). These results lead to the intriguing conclusion that the PPC plays a causal role in shaping the intrinsic neuronal dynamics and functional properties of the FEF.

The methodology employed in this study is rigorous, well-suited to the research questions, and clearly described. However, several of the claims made in the manuscript lack sufficient support, as outlined in the points below. These shortcomings somewhat undermine the overall persuasiveness of the work and suggest that the manuscript might be better positioned for publication in a more specialized neuroscience journal.

Major points:

1. PPC inactivation determines the PPC input to FEF neurons.

The authors interpret the absence of modulation in intrinsic timescales during PPC inactivation experiments (Figs. 4 and 5) as evidence that such modulation depends on input from the PPC. Since the FEF receives direct projections not only from the PPC but also from other posterior visual areas, including regions in the occipital lobes, the authors note that "PPC inactivation significantly altered their intrinsic dynamics but had a limited impact on their visual responses, possibly due to compensatory inputs from other visual regions." This suggests that inactivating the PPC slows down neuronal dynamics in the FEF, potentially by unmasking slower local dynamics or by enhancing activity within local recurrent circuits. However, because PPC and FEF are components of a broader network, the authors should clarify that it is an oversimplification to attribute changes solely to PPC input to the FEF. In other words, the causal relationship between PPC and FEF activity is likely indirect and mediated by multiple interconnected pathways. These considerations imply that the circuit mechanism proposed in Figure 6 simplifies the underlying processes occurring during PPC inactivation and does not fully capture the complexity of network interactions involved.

2. Behavioral effects and functional relevance of intrinsic timescales.

At line 210 the authors state that PPC inactivation produced "robust behavioral effects" in both the free viewing and double-target choice tasks, citing Chen et al. (2020). However, the free viewing task, as described in Chen et al. (2020), appears to involve only passive observation of visual stimuli (Fig. 1B). If no active behavioral response (e.g., saccades, decisions) was required, what measurable "behavioral effects" were observed during inactivation? Clarifying whether these effects relate to eye movements, fixation patterns, or other quantifiable metrics is essential to establish the functional relevance of intrinsic timescales.

More specifically, while Fig. 3A demonstrates a correlation between intrinsic timescales and neuronal activity modulations (single/popout indexes), the functional relevance of this relationship remains unclear in the absence of behavioral changes

during PPC inactivation. If neuronal changes (e.g., altered single/popout index correlations) occurred without any modulation of the behavioral reports, this weakens the argument that intrinsic timescales are behaviorally meaningful in this context.

The manuscript should explicitly acknowledge that the dissociation between neuronal and behavioral changes during inactivation complicates inferences about the purpose or necessity of intrinsic timescales. For example, are these timescales merely epiphenomenal in passive tasks, or do they reflect latent processes not captured by the current behavioral measures? Addressing this ambiguity would prevent overinterpretation of the results.

3. Why pool MUA and SUA together?

At line 99, it is clearly stated that the analyzed dataset consists of "400 single- and multi-unit responses." The authors chose to pool these heterogeneous neuronal signals together without providing any justification. Since MUA reflects the combined spiking of multiple neurons near the electrode tip, its spike-count autocorrelation likely represents an average across contributing cells. Consequently, timescales derived from MUA might tend to be longer than those computed from SUA autocorrelations. This difference may arise because averaging smooths out fast fluctuations in spiking activity.

If this is the case, the observed bimodal distribution of timescales might simply reflect the distinct contributions of MUA and SUA, with MUA associated with longer timescales and SUA with shorter ones. The authors should therefore explicitly examine how the timescale distributions differ between MUA and SUA populations to clarify this point.

4. Are bimodal distributions of intrinsic timescales novel?

A very recent publication by Zeisler et al. (J Neurosci 2025, DOI: 10.1523/JNEUROSCI.2155-24.2025), which is not cited in the manuscript, provides compelling evidence that bimodal distributions of neuronal timescales are a conserved feature across multiple species and brain regions. Such bimodality seems then to be a general organizational principle rather than a region- or species-specific phenomenon. Although Zeisler et al. do not specifically investigate the FEF, their findings are highly relevant to the current study's claim with the consequence that the observation of bimodal timescales in FEF, while still valuable, appears less unexpected or novel. The manuscript would benefit from acknowledging this contemporary work and discussing how the present findings in FEF fit within the emerging understanding that bimodal timescale distributions may be a widespread feature of cortical organization.

5. PPC inactivation affects the two monkeys differently.

At line 220, the authors state that "This increase in intrinsic neural timescales was evident in both monkeys." However, a close inspection of Figure 4B reveals that most data points (i.e., neurons) lie above the dashed line for Monkey Q, whereas for Monkey J, the majority appear below it, if I am not wrong. To verify this apparent discrepancy, I recommend computing separate histograms of the differences $\log(\tau_{\text{inact}}) - \log(\tau_{\text{control}})$ for each monkey. This analysis would allow assessment of whether the median changes are significantly positive for one animal and negative for the other, thereby clarifying the differential effects of PPC inactivation.

(Remarks on code availability)

Version 1:

Reviewer comments:

Reviewer #1

(Remarks to the Author)

I have no further comments. The authors have satisfactorily addressed my previous concerns.

(Remarks on code availability)

Reviewer #2

(Remarks to the Author)

The authors have satisfactorily addressed all my concerns, and I endorse the paper for publication. A minor note: panel B in Figure S2 seems unnecessary, as correlating half of the data with the full dataset will inherently produce a strong correlation.

(Remarks on code availability)

Reviewer #3

(Remarks to the Author)

All concerns from my previous review are adequately addressed; I have no further comments, and can recommend the manuscript for publication.

(Remarks on code availability)

Reviewer #4

(Remarks to the Author)

The revised version of the manuscript has been significantly improved, and the authors have convincingly addressed all my concerns.

More specifically:

- Regarding Point 1: The authors did not change the Fig. 6 but better refined the caption emphasizing in the Discussion that it serves only as a schematic illustration of how a broad fronto-parietal network can reciprocally modulate the timescales expressed by FEF neurons. This is a sufficient effort as I mainly asked to clarify that the inactivation of PPC can have widespread effects and only indirectly influence FEF. This point is now clearly stated in the Discussion.
- Regarding Point 2: The added text in the Discussion now clarifies the extent to which the relationship between neural activity and behavioral modulation is addressed in this work, thereby solving the issue of overinterpretation of the presented results.
- Regarding Point 3: The authors added a new supplementary figure that convincingly addresses my concern. A minor point to be considered at this stage is to provide the parameters used for the kernel density estimates in the caption of Fig. S4 – for SUA (left) and MUA (right) – as they are likely different.
- Regarding Point 4: The authors have included the requested citation and added a clarifying text in the Discussion fully addressing my criticism.
- Regarding Point 5: The Supplementary Fig. S10 convincingly demonstrates that, despite my concerns, PPC inactivation increases intrinsic neural timescales in both monkeys.

(Remarks on code availability)

We thank all reviewers for their thoughtful and constructive comments. Our point-by-point responses to each of the reviewers' comments are below.

Black bold: Reviewer comment

Red regular: Our response

Red italics: Excerpt from the revised manuscript

Reviewer Comments

Reviewer #1

In this manuscript, Soyuhos et al. have investigated intrinsic timescales in the frontal eye field (FEF), an area that plays a key role in stimulus driven and goal directed visual attention. They identified two distinct neuronal populations within FEF — one with short and another one with long intrinsic timescales — that contribute differentially to transient stimulus processing and attentional functions, respectively. To test whether these timescales are influenced by factors beyond local circuitry, the authors inactivated the posterior parietal cortex (PPC) using cryoloops implanted in the intraparietal sulcus (IPS). This manipulation significantly reduced the number of short-timescale neurons and increased the number of long-timescale neurons in FEF, resulting in an overall shift toward longer intrinsic timescales. Additionally, PPC inactivation selectively disrupted the attentional modulation in long τ neurons. These findings highlight the importance of interareal interactions, demonstrating that intrinsic timescales are not solely shaped by local network interactions but also by input from other brain regions.

While the study provides important insights on the heterogeneity of time intrinsic time scales with FEF, it lacks important large-scale population and laminar analysis. Such data and analysis could potentially offer a more comprehensive understanding of the spatial distribution and population-level dynamics/computations associated with the distinct neuronal classes identified in this study.

Regarding novelty and significance, previous studies have already reported the heterogeneity in time scales within cortical areas, and those studies are cited. They have now shown that this also holds for FEF, an area important for visual attention. Their second finding, the influence of PPC on intrinsic timescales in FEF, is the real new finding. However, one cannot rule out the effect of other brain areas that might have contributed to the change in time scales with the inactivation of PPC. Hence the changes they report might not be exclusively determined by PPC.

We thank the reviewer for their helpful guidance and constructive feedback.

1. Do the two distinct neuron types based on intrinsic timescales form spatially segregated populations, and is there higher/lower functional coupling within/between these populations? Additionally, I am wondering whether there is a laminar dependence of the location of these distinct classes of neurons which could be important given how the input from PPC modulates their dynamics. Given that the authors recorded using laminar probes, in principle it should be possible to extract that information from the data.

We thank the reviewer for this suggestion. Although laminar analyses in the FEF are challenging due to the cortical folding, we performed an additional analysis examining the recording depth of each neuron. This analysis revealed a clear spatial segregation between the two neuronal classes. Short- τ neurons were located

at more superficial contacts (median depth = 901 μm), whereas long- τ neurons were predominantly found at deeper contacts (median depth = 1351 μm). These results strengthen our claim of two distinct neuronal populations and suggest that they may preferentially be located at different cortical layers. The new findings are now presented in Figure 2E and Supplementary Figure S3, and described in the Results section.

Page 3; lines 115-122: “... In addition, we observed a significant depth-dependent separation between the two neuronal classes. Short- τ neurons were recorded at more superficial sites ($N = 151$; median depth = 901 μm), whereas long- τ neurons were found deeper ($N = 199$; median depth = 1351 μm ; Wilcoxon rank-sum test, $p < 0.001$; Fig. 2E). The likelihood of encountering a long- τ neuron increased with depth (channels 8–16; 1050–2250 μm ; Wilcoxon signed-rank test, $p = 3.59 \times 10^{-2}$; Fig. S3). Although absolute recording depth does not directly correspond to cortical layers due to the cortical fold and relative penetration angle, these results reveal a systematic depth-dependent organization of neurons with distinct intrinsic timescales.”

Regarding functional coupling, we performed a preliminary analysis of noise correlations and found that pairs of long- τ neurons showed significantly higher correlations compared to pairs of short- τ neurons or mixed short-long pairs. We have included a figure in this response letter (see Rebuttal Figure 1) to share this finding directly. While these results are interesting and align with the idea of distinct functional ensembles, a full exploration of these network-level interactions is the focus of a follow-up project. Therefore, to keep the manuscript focused, we have not included this analysis in the current revision. We are grateful for the suggestion, which confirms an important direction for our future research.

2. In the decoding analysis, the decoder performance is already highly saturated for the classes used (in/out of RF). Wouldn't it be more appropriate to use a set of stimuli try to decode stimulus information from neural responses of these 2 groups of neurons?

We thank the reviewer for this suggestion. We have now performed an additional, more demanding decoding analysis. In addition to the original binary 'detectability' task (In-RF vs. Out-RF), we implemented a multiclass discriminability analysis to decode the stimulus location across 24 possible positions. The results of this more stringent multiclass analysis corroborate and extend our original findings, further confirming the robustness of the decoding results. These new analyses are now reported in the main text.

Page 4; lines 172-196: “... Next, we assessed the visual and salience information encoded at the population level by short- τ and long- τ neurons. We performed two complementary pseudo-population decoding analyses. The first was a binary stimulus detectability task, in which we predicted whether a stimulus location was inside or outside the receptive field. The second was a multi-class stimulus discriminability task, where we predicted the specific location of the stimulus out of 24 possibilities. We then generated neuron-dropping curves (NDCs) from neuronal subsets of each group to examine decoding accuracy across varying population sizes (Fig. S9 and Fig. 3D-E). To quantify the performance of each population, we calculated the area under these curves (AUC) that exceeded the chance level.

In the detectability task, we found a dissociation between the two populations (Fig. S9). For single-stimulus detection, short- τ neurons excelled during the transient period (Wilcoxon rank-sum test: $p = 2.41 \times 10^{-13}$, effect size = 0.60; median = 0.989 for short- τ vs. 0.984 for long- τ), whereas long- τ neurons performed better during the sustained period ($p = 7.71 \times 10^{-3}$, effect size = 0.22; median = 0.953 for long- τ vs. 0.947 for short- τ). For popout detection, long- τ neurons consistently showed stronger performance in both transient ($p = 3.94 \times 10^{-4}$, effect size = 0.29; median = 0.64 for long- τ vs. 0.62 for short- τ) and sustained periods ($p = 1.88 \times 10^{-7}$, effect size = 0.43; median = 0.76 for long- τ vs. 0.73 for short- τ).

This pattern of specialization was even more evident in the discriminability task (Fig. 3F-G). The single-stimulus discrimination results mirrored the detectability results: short- τ neurons again had better performance in the transient period ($p = 2.56 \times 10^{-36}$, effect size = 1.00; median = 0.31 for short- τ vs. 0.27 for

long- τ), while long- τ neurons were better during the sustained period ($p = 4.88 \times 10^{-16}$, effect size = 0.64; median = 0.18 for long- τ vs. 0.17 for short- τ). This advantage for long- τ neurons was most apparent for popout discrimination, where they demonstrated better performance across both transient ($p = 1.44 \times 10^{-16}$, effect size = 0.68; median = 0.058 for long- τ vs. 0.054 for short- τ) and sustained periods ($p = 2.56 \times 10^{-34}$, effect size = 1.00; median = 0.09 for long- τ vs. 0.06 for short- τ). The differences between the groups were most pronounced for single-stimulus discrimination in the transient period and popout stimulus discrimination in the sustained period.”

3. The authors should include representative autocorrelograms for the two neuronal classes in Figure 1 as it will enhance the reader’s understanding of intrinsic timescales by providing visual intuition

We thank the reviewer for this helpful suggestion. We agree that this visualization is very useful. We have now added a new panel to Figure 2 (Panel C) showing representative autocorrelograms and their exponential fits for both a short- τ and a long- τ neuron, providing a clear visual intuition for timescale measurement.

4. The authors mention that the FEF receives inputs from multiple brain areas. In this study they inactivated one of its inputs from the PPC. Do the authors have any mechanistic understanding on how inactivation of other brain areas might influence intrinsic timescales, and how these effects might compare in magnitude to the PPC inactivation results?

We thank the reviewer for raising this important point. Beyond the PPC, the FEF receives convergent inputs from multiple extrastriate visual areas, including from the pulvinar (Arcaro et al., 2018). We have expanded the Discussion to address two considerations: First, anatomical studies indicate that the reciprocal projections between PPC (particularly LIP) and FEF are especially strong, comprising a major fraction of its long-range visual inputs (Schall et al., 1995; Stanton et al., 1995). Given this dominant connectivity, PPC inactivation is likely to exert a more substantial influence on FEF dynamics than perturbations of other individual inputs. Second, we propose two complementary mechanisms for the observed slowing of timescales: 1) Because parietal neurons exhibit faster dynamics than prefrontal neurons (Chaudhuri et al., 2015; Murray et al., 2014; Wasmuht et al., 2018), removing this fast PPC input may unmask slower local dynamics within FEF, particularly in short- τ neurons with weaker recurrent connectivity. 2) PPC inactivation may also enhance local recurrent engagement within FEF or broader prefrontal loops, further prolonging integration times (Litwin-Kumar & Doiron, 2012). Based on this framework, we predict that removal of inputs from areas with shorter intrinsic dynamics (e.g., parietal cortex, occipital visual areas) would similarly prolong FEF timescales, whereas inactivating higher-order frontal inputs with longer dynamics (e.g., supplementary eye field) might instead shorten FEF timescales by reducing recurrent drive.

We have added this discussion to the revised manuscript (Page 8, lines 368-371). While these predictions remain to be directly tested, they provide a reference for future studies aimed at dissecting how distinct long-range inputs shape the temporal dynamics of prefrontal circuits.

5. Is there a possibility that inactivating PPC might affect other brain areas which in turn feed into FEF? As a result, can the findings reported here be the combined effect of indirect perturbation of other areas in addition to PPC? These possibilities are not adequately considered, which weaken the authors’ conclusions.

We thank the reviewer for raising this important point. We completely agree that the PPC and FEF are nodes within a large-scale, interconnected network and that attributing the observed effects solely to a direct, monosynaptic PPC-FEF projection would be an oversimplification. For example, pulvinar may serve as an important cortical-thalamic pathway regulating dynamics in both PPC and FEF (Aussel et al., 2023; Boshra & Kastner, 2022; Fiebelkorn & Kastner, 2019). Our results demonstrate that PPC activity is causally necessary

for normal FEF dynamics; its removal perturbs the FEF in a specific manner, regardless of whether the influence is direct or mediated by polysynaptic pathways through other brain regions. To address this point more explicitly, we have revised the Discussion (Page 8, lines 355–361; Page 22, line 708) to acknowledge that indirect effects cannot be ruled out and have framed our conclusions more cautiously in the context of broader network interactions.

Reviewer #2

In this study, Syuhos et al. investigate how intrinsic neural timescales vary across neurons in the frontal eye field (FEF), considering both stimulus-driven and attention-modulated responses, and how these timescales are affected by inactivation of the posterior parietal cortex (PPC)—a region known to regulate attentional signals in FEF. Intrinsic timescales were estimated via the autocorrelation decay of spike-count fluctuations during spontaneous activity (i.e., the baseline period preceding stimulus onset).

The authors identified two distinct classes of FEF neurons based on their intrinsic timescales: short (<50 ms) and long (50–150 ms). When analyzing the neurons' responses to visual stimuli, they found that short-timescale neurons were more active during the transient, early phase of the response, whereas long-timescale neurons exhibited stronger attentional modulation during the later, sustained phase. PPC inactivation led to an overall increase in FEF timescales, with a stronger effect observed in short-timescale neurons. Notably, attentional modulation was reduced following PPC inactivation, especially in long-timescale neurons.

I found this study compelling. It is clearly written, the methods are well described, and the results are appropriately analyzed and interpreted. Nonetheless, I have a few concerns regarding the stability of the intrinsic timescale estimates and the theoretical framing of the findings:

We thank the reviewer for their detailed feedback and helpful suggestions.

1. Stability of intrinsic timescales: The robustness of the estimated timescales should be assessed. For instance, the authors could divide each recording session into two halves and compute timescales separately to test their consistency (e.g., via a split-half correlation). This would help ensure that the timescale classification reflects stable intrinsic properties rather than session-specific noise.

We thank the reviewer for this suggestion. As recommended, we have now performed a split-half analysis by calculating intrinsic timescales separately for the first and second halves of each recording session. The analysis revealed a strong, significant positive correlation between the two halves (Spearman's $\rho = 0.72$, $p < 0.001$), confirming that our timescale estimates are robust and reflect stable intrinsic properties of the neurons. This new analysis is presented in the new Supplementary Figure S2 and described in the Results section.

Page 3; lines 102-105: "... Furthermore, a split-half analysis confirmed the stability of the measurements, showing strong positive correlations both between the two halves (Spearman's $\rho = 0.72$) and of each half with the full session data ($\rho = 0.85$ and 0.88 , respectively; $p < 0.001$; Fig. S2)."

2. Theoretical framework: The study would benefit from a broader theoretical context. Recent work (Ponce-Alvarez et al., 2025, DOI: 10.1523/JNEUROSCI.1699-24.2024) has shown that neural timescales and variances relate systematically to a network's topology and functional roles. Specifically, neurons at the core of the network exhibit longer timescales, lower variability, and weaker stimulus responses, while peripheral neurons show shorter timescales and greater responsiveness. This reflects a

fundamental trade-off between stability and flexibility in network dynamics. Framing the current results within this context could strengthen the interpretation of the timescale classes and their functional relevance.

We thank the reviewer for this suggestion and for pointing us to the highly relevant work of Ponce-Alvarez et al. (2025). We agree that this theoretical framework greatly enriches the interpretation of our findings. We have now revised our Discussion to incorporate this "core-periphery" context. Specifically, we discuss how our two neuronal classes align with this framework: our short- τ neurons, with their higher variability and transient responses, match the functional properties of a flexible "periphery," while our long- τ neurons, characterized by sustained activity, are consistent with a stable "core." These additions can be found in the revised Discussion section (Page 7, lines 299–304).

3. Additional analyses: Following Ponce-Alvarez et al. (2025), it would be informative to analyze the intrinsic variance and pairwise correlations among neurons (and how they change during PPC inactivation). One might expect short-timescale neurons to exhibit higher variability and weaker correlations, whereas long-timescale neurons may show lower variance and stronger correlations among them. Exploring these dimensions could provide a more comprehensive characterization of the functional organization of the FEF.

We thank the reviewer for these suggestions. We have performed both of the requested analyses, and the results strongly support the reviewer's predictions and the core-periphery framework. First, we analyzed the temporal variability of baseline firing rates and found that short- τ neurons exhibit significantly higher variability than long- τ neurons. This new analysis is presented in Supplementary Figure S6 and described in the Results section.

Page 3; lines 129-134: "... However, the two groups differed in their intrinsic firing patterns. Short- τ neurons exhibited significantly higher temporal variability (Wilcoxon rank-sum test: $p = 4.6 \times 10^{-5}$; Fig. S6A) but similar trial-to-trial variability ($p = 0.31$; Fig. S6B) in their firing rate compared to long- τ neurons. This suggests that a short- τ signifies more transient activity characterized by a dynamic baseline firing pattern, whereas a long- τ indicates sustained activity with a stable baseline firing pattern."

Additionally, we analyzed pairwise noise correlations and confirmed that pairs of long- τ neurons are indeed more strongly correlated. As this second analysis is part of a larger, ongoing project, we have not included it in the manuscript itself but have shared the result in this letter for the reviewer's consideration (see Rebuttal Figure 1). We are grateful for these suggestions, which have strengthened our manuscript and validated a key aspect of our future work.

Other comments:

- It seems that some neurons displayed significantly low autocorrelation values at short time lags, which led the authors to fit exponential functions to the autocorrelation functions (ACFs) for non-zero lags. Are these particular neurons classified as belonging to short- or long-timescale group?

Our analysis confirms that the properties of the initial dip in the ACF do not systematically differentiate the short- τ and long- τ groups. Specifically, we compared both the magnitude and timing of the initial ACF profile. We found no significant difference in the magnitude at the first time lag ($p = 0.31$) or at the subsequent peak ($p = 0.17$). We have now added this analysis to the Methods section.

Page 10; lines 436-438: “... Here, the initial dip in autocorrelation profiles did not systematically differ between short- τ and long- τ groups; neither the magnitude at the first time lag ($p = 0.31$) nor at the subsequent peak ($p = 0.17$) differed significantly (Wilcoxon rank-sum test).”

- The uncertainty of the estimated ACF decay (τ) should be reported. Additionally, it would be informative to examine how this uncertainty varies as a function of the timescale.

We now report the uncertainty of our timescale estimates using the coefficient of variation (CV). The median CV was 0.22, indicating high precision. This is now described in the Methods and Results section.

Page 3; lines 100-105: “... Based on this, neurons with poor fits ($R^2 < 0.3$, $n = 20$) were excluded, leaving 380 neurons for analysis. For this final set of neurons, the precision of the fits was high with a median coefficient of variation (CV) of 0.22.”

Page 3; lines 124-127: “... Consistent with this finding, the distinction between timescale groups was not explained by unit type (Chi-squared test of independence: $\chi^2(1) = 1.38$, $p = 0.24$) (Fig. S4) or by differences in the precision of the timescale estimates (Wilcoxon rank-sum test: $p = 0.56$; median CV = 0.21 for short- τ vs. 0.24 for long- τ).”

- Figures 5C and 5D are difficult to interpret and could be improved for clarity. In particular, the use of a secondary axis should be avoided, as it is confusing.

We thank the reviewer for this feedback. To address this, we have redesigned these panels to be more direct and intuitive. The new Figures 5C and 5D now directly plot the relative chance in decoding accuracy as a function of population size, which clearly visualizes the magnitude of the change without needing a secondary axis. For readers interested in the full performance curves for both conditions, we have moved the original plots to the Supplementary Figure S13. This revised presentation makes the results much clearer and easier to interpret.

- I could not find the number of trials for each condition.

We apologize for this omission. We have now added the average number of trials per condition to the Methods section.

Page 9; lines 392-393: “... On average, there were 20 trials per location for each stimulus type in both the control and inactivation conditions.”

- For the decoding, take into account the confidence interval of the chance level (which depends on the number of trials).

We thank the reviewer for this suggestion. Our original analysis used a fixed chance level as the baseline. We agree that a more rigorous approach is to use a baseline that accounts for the statistical confidence interval of chance-level performance, which varies with the number of test trials available at each population size. Following the reviewer’s advice, we have updated our analysis pipeline. For each step in the neuron-dropping curve (i.e., for each number of neurons), we now compute the 95% confidence interval around the chance level using a binomial distribution. The number of trials used for this calculation is determined by simulating our neuron and trial sampling procedure, which accurately reflects the number of test trials available to the decoder at that specific population size. This results in a baseline that correctly captures the statistical uncertainty

across the analysis. Our primary metric, the Area Under the Curve (AUC), is now calculated as the area between the decoding accuracy curve and the upper bound of this CI. We have re-run our analysis with this improved method and confirmed that our initial findings remain the same. We have updated the Methods section to reflect this improved analysis.

Page 13; lines 562-574: "... First, a statistical baseline was established by computing the 95% confidence interval (CI) of chance performance for each neuron count. This CI was calculated from the 2.5th and 97.5th percentiles of a binomial distribution, with the number of trials at each population size estimated by simulating the neuron and trial sampling procedure 1000 times. For the detectability task, the chance probability was $p = 0.5$, whereas for the discriminability task, it was $p = 1/24$, where 24 is the number of stimulus locations. The AUC was then calculated as the area between the decoding accuracy curve and the upper bound of this chance-level CI, using the trapezoidal rule. To generate a distribution of AUC scores, the 500 resampling iterations were randomly partitioned into 100 non-overlapping groups, and the AUC was computed for the average decoding curve of each group. These raw AUC values were then normalized by dividing each by the maximum possible area above the chance CI (i.e., the area between a perfect 100% accuracy curve and the CI's upper bound). This yields a normalized score from 0 to 1, where 0 represents performance indistinguishable from chance and 1 represents perfect decoding across all population sizes."

I hope the authors find these comments constructive.

Reviewer #2 (Remarks on code availability):

The legibility of the provided code could be improved, although it appears to contain all the necessary information to reproduce the results. A README file is included with sufficient instructions for installing and running the application. I did not attempt to install or run the code.

We thank the reviewer for their feedback on the code. We have now substantially modularized the code to improve its organization. The various functions used in the main script have been grouped into separate folders corresponding to their respective analyses. Furthermore, each function is now organized into a stand-alone script that includes a detailed docstring explaining its purpose and input variables. The main README file has also been updated to reflect this improved folder structure and provide a clearer guide for navigating the code.

Reviewer #3

The authors demonstrate that within the same cortical area, the frontal eye field, there are populations of neurons operating at a shorter and a longer time scale; these populations have distinct functional properties, and are differentially affected by inactivation of the posterior parietal cortex.

Experiments are clearly designed; results are solid and clearly presented.

We thank the reviewer for their kind words.

I have only minor comments/suggestions.

Species might be indicated in the title or abstract

We thank the reviewer for their suggestion. We have now specified the species in the Abstract.

Page 1; line 13: "... We measured intrinsic timescales of frontal eye field (FEF) neurons in rhesus macaques and examined their changes during posterior parietal cortex (PPC) inactivation."

Fig 1 legend

or the same stimulus embedded among identically colored distractors ("Popout")

-> distractors of a different uniform color (or other wording? now, "identically colored" may refer to either stimulus or distractors).

We thank the reviewer for pointing out this ambiguity. We have revised the legend for Figure 1B.

Page 15; lines 633-635: "... (B) Visual stimulus conditions included either a single colored stimulus presented alone ("Single") or the same stimulus embedded among distractors of a different, uniform color ("Popout")."

Fig. 1 C,D - time bin 10 ms?

We have now specified the time bin (10 ms) in the legends for Figure 1C and Figure 1D.

Page 15; lines 637 and 639: "... (C) Example neuronal response illustrating mean firing rate to a single stimulus presented either inside (blue trace) or outside (gray trace) the neuron's receptive field (10 ms bins). (D) Intrinsic neural timescales were measured using spike-count autocorrelation computed from neuronal activity during the baseline period. Blue dots indicate autocorrelation coefficients at successive 10 ms time lags."

Fig 6. Legend:

Short-timescale (short- τ) neurons in the frontal eye field (FEF; blue) receive direct feedforward inputs from the posterior parietal cortex (PPC; orange) and early visual cortical areas (pink).

in the scheme, there is no direct Visual -> FEF connection; only via PPC.

We thank the reviewer for this suggestion. We have revised the schematic in Figure 6A and its legend to clearly illustrate and describe the direct feedforward connection from early visual areas to the FEF (Page 22).

Reviewer #3 (Remarks on code availability):

code will be made publicly available after acceptance of the ms

Reviewer #4

In this study, the authors present a novel analysis of a dataset originally collected by Chen et al. (Neuron, 2020), in which two monkeys performed visual sensitivity and saliency-driven attention tasks under both PPC-inactivation and control conditions. Neural activity was recorded in the frontal eye fields (FEF), revealing that neurons with shorter intrinsic timescales exhibited stronger transient responses to visual stimuli, suggesting a role in rapid visual processing. In contrast, neurons with longer intrinsic timescales showed more sustained responses, indicating their involvement in maintaining attention over time.

Importantly, PPC inactivation caused a lengthening of the shorter intrinsic timescales and reduced attentional modulation in neurons with longer timescales, considering both multi-unit and single-unit activity (namely, MUA and SUA respectively). These results lead to the intriguing conclusion that the PPC plays a causal role in shaping the intrinsic neuronal dynamics and functional properties of the FEF.

The methodology employed in this study is rigorous, well-suited to the research questions, and clearly described. However, several of the claims made in the manuscript lack sufficient support, as outlined in the points below. These shortcomings somewhat undermine the overall persuasiveness of the work and suggest that the manuscript might be better positioned for publication in a more specialized neuroscience journal.

We thank the reviewer for their valuable feedback on our manuscript.

Major points:

1. PPC inactivation determines the PPC input to FEF neurons.

The authors interpret the absence of modulation in intrinsic timescales during PPC inactivation experiments (Figs. 4 and 5) as evidence that such modulation depends on input from the PPC. Since the FEF receives direct projections not only from the PPC but also from other posterior visual areas, including regions in the occipital lobes, the authors note that "PPC inactivation significantly altered their intrinsic dynamics but had a limited impact on their visual responses, possibly due to compensatory inputs from other visual regions." This suggests that inactivating the PPC slows down neuronal dynamics in the FEF, potentially by unmasking slower local dynamics or by enhancing activity within local recurrent circuits.

However, because PPC and FEF are components of a broader network, the authors should clarify that it is an oversimplification to attribute changes solely to PPC input to the FEF. In other words, the causal relationship between PPC and FEF activity is likely indirect and mediated by multiple interconnected pathways. These considerations imply that the circuit mechanism proposed in Figure 6 simplifies the underlying processes occurring during PPC inactivation and does not fully capture the complexity of network interactions involved.

We thank the reviewer for raising this point. We completely agree that the PPC and FEF are nodes within a large-scale network and that attributing our results solely to a direct, monosynaptic projection would be an oversimplification. Our goal was to establish a causal link showing that PPC activity is necessary for normal FEF dynamics, regardless of whether the influence is direct or mediated by polysynaptic pathways. To address this point explicitly, we have revised the Discussion to acknowledge that indirect effects cannot be ruled out and have framed our conclusions more cautiously (Page 8, lines 355-361). To further ensure clarity for the reader, we have also amended the text to emphasize that our schematic in Figure 6 is intended as a functional model of the proposed interactions, rather than a complete anatomical circuit diagram (Page 7, line 322; Page 22, lines 705 and 714-716).

2. Behavioral effects and functional relevance of intrinsic timescales.

At line 210 the authors state that PPC inactivation produced "robust behavioral effects" in both the free viewing and double-target choice tasks, citing Chen et al. (2020). However, the free viewing task, as described in Chen et al. (2020), appears to involve only passive observation of visual stimuli (Fig. 1B). If no active behavioral response (e.g., saccades, decisions) was required, what measurable "behavioral effects" were observed during inactivation? Clarifying whether these effects relate to eye movements, fixation patterns, or other quantifiable metrics is essential to establish the functional relevance of intrinsic timescales.

More specifically, while Fig. 3A demonstrates a correlation between intrinsic timescales and neuronal activity modulations (single/popout indexes), the functional relevance of this relationship remains unclear in the absence of behavioral changes during PPC inactivation. If neuronal changes (e.g., altered single/popout index correlations) occurred without any modulation of the behavioral reports, this weakens the argument that intrinsic timescales are behaviorally meaningful in this context.

The manuscript should explicitly acknowledge that the dissociation between neuronal and behavioral changes during inactivation complicates inferences about the purpose or necessity of intrinsic timescales. For example, are these timescales merely epiphenomenal in passive tasks, or do they reflect latent processes not captured by the current behavioral measures? Addressing this ambiguity would prevent overinterpretation of the results.

We thank the reviewer for this critical and detailed comment. It gives us the opportunity to clarify the behavioral findings from our prior work, the rationale for our current task design, and how we have now strengthened the discussion of our findings in the manuscript.

First, we would like to clarify a key point about the tasks cited from Chen et al. (2020). While the task in our current manuscript is passive, the "free viewing" task described in Chen et al. was, in fact, an active behavioral task where the same monkeys in the current study made saccades to freely explore hundreds of natural images. The "robust behavioral effect" we refer to was a significant and quantifiable impairment: during PPC inactivation, the monkeys' eye movements were no longer properly guided by the visual salience of the image in both the contralateral field and contralaterally directed saccades. This establishes a direct, causal link between PPC activity and the motor behavior of saccadic guidance to salient stimuli. In addition to the free-viewing paradigm, we also measured the change in target selection in the onset synchrony task.

The reason we then used a passive design in the current study was to isolate the underlying neural mechanism. Our goal was to characterize the representation of stimulus salience in the FEF. However, an active saccade task would have introduced confounding motor-planning signals and movement-related modulation, which are known to heavily modulate FEF activity (Bruce & Goldberg, 1985; Hanes & Schall, 1996). By using a passive design, we could show that PPC inactivation specifically disrupts these salience signals in FEF. Importantly, in the present work, the neurophysiology was recorded in the same animals and within overlapping experimental sessions that included the behavioral free-viewing paradigm and the onset synchrony task. These observed neural changes suggest a mechanistic explanation for the behavioral impairments observed in the active tasks.

Finally, we agree with the reviewer that the manuscript should explicitly acknowledge the nuances of interpreting these neural changes. Following their suggestion, we have revised the final paragraphs of the Discussion (Page 8; lines 341-353).

3. Why pool MUA and SUA together?

At line 99, it is clearly stated that the analyzed dataset consists of "400 single- and multi-unit responses." The authors chose to pool these heterogeneous neuronal signals together without providing any justification. Since MUA reflects the combined spiking of multiple neurons near the electrode tip, its spike-count autocorrelation likely represents an average across contributing cells. Consequently, timescales derived from MUA might tend to be longer than those computed from SUA autocorrelations. This difference may arise because averaging smooths out fast fluctuations in spiking activity.

If this is the case, the observed bimodal distribution of timescales might simply reflect the distinct contributions of MUA and SUA, with MUA associated with longer timescales and SUA with shorter ones. The authors should therefore explicitly examine how the timescale distributions differ between MUA and SUA populations to clarify this point.

We thank the reviewer for raising this important point. We have now analyzed the timescale distributions for single-unit (SUA) and multi-unit (MUA) populations independently. Our results show that the bimodal distribution of intrinsic timescales is present in both populations. Specifically, the distribution was significantly bimodal for both SUA (Excess mass test: $p = 2 \times 10^{-3}$) and MUA ($p < 2.2 \times 10^{-16}$) recordings. Furthermore, a chi-squared test of independence confirmed that there is no significant association between unit type and

timescale class ($\chi^2(1) = 1.38, p=0.24$). We have added a new supplementary figure (Fig. S4) to visualize these separate distributions and have updated the Results section accordingly.

Page 3; lines 123-127: “... *This bimodal timescale distribution was evident within both the multi-unit ($N = 300$; Excess mass test: $p < 2.2 \times 10^{-16}$) and single-unit ($N = 80$; $p = 2 \times 10^{-3}$) populations. Consistent with this finding, the distinction between timescale groups was not explained by unit type (Chi-squared test of independence: $\chi^2(1) = 1.38, p = 0.24$) (Fig. S4) ...*”

4. Are bimodal distributions of intrinsic timescales novel?

A very recent publication by Zeisler et al. (J Neurosci 2025, DOI: 10.1523/JNEUROSCI.2155-24.2025), which is not cited in the manuscript, provides compelling evidence that bimodal distributions of neuronal timescales are a conserved feature across multiple species and brain regions. Such bimodality seems then to be a general organizational principle rather than a region- or species-specific phenomenon. Although Zeisler et al. do not specifically investigate the FEF, their findings are highly relevant to the current study’s claim with the consequence that the observation of bimodal timescales in FEF, while still valuable, appears less unexpected or novel. The manuscript would benefit from acknowledging this contemporary work and discussing how the present findings in FEF fit within the emerging understanding that bimodal timescale distributions may be a widespread feature of cortical organization.

We thank the reviewer for bringing this very recent study to our attention. We have now cited this important work and revised our Discussion to place our findings within this broader context. We agree that our work contributes to the emerging view that timescale heterogeneity is a conserved organizational principle. Our study, however, provides three critical and novel advances that build upon this foundation. We first establish this principle in the FEF, a key cortical hub for attention not previously studied in this context. Furthermore, our primary contribution is providing a functional dissociation between these populations, linking short-timescale neurons to rapid visual processing and long-timescale neurons to salience representation. Finally, we offer the first causal evidence that these intrinsic timescales are not fixed but are actively modulated by long-range cortical inputs from the PPC. We therefore believe our study provides a mechanistic understanding and causal extension to the general principle they describe. This view is supported by Zeisler et al. (2025) themselves, who cite our preprint when discussing the need for such functional characterization.

Page 7; lines 294-299: “... *This finding aligns with recent reports of multimodal timescale distributions across the cortex in rodents, macaques, and humans (Zeisler et al., 2025; Shi et al., 2025). Here, we provide the first evidence for this dual-timescale organization in the FEF, a critical prefrontal oculomotor area in the frontoparietal attention network (Boshra & Kastner, 2022; Xia et al., 2024), and show that these two classes support a functional division of labor.*”

5. PPC inactivation affects the two monkeys differently.

At line 220, the authors state that “This increase in intrinsic neural timescales was evident in both monkeys.” However, a close inspection of Figure 4B reveals that most data points (i.e., neurons) lie above the dashed line for Monkey Q, whereas for Monkey J, the majority appear below it, if I am not wrong. To verify this apparent discrepancy, I recommend computing separate histograms of the differences $\log(\tau_{\text{inact}}) - \log(\tau_{\text{control}})$ for each monkey. This analysis would allow assessment of whether the median changes are significantly positive for one animal and negative for the other, thereby clarifying the differential effects of PPC inactivation.

We thank the reviewer for their comment. While our original submission included statistics showing a significant effect in each animal and differences in their effect sizes, we agree with the reviewer that a visual

representation for each monkey makes the results clearer. We have now created a new figure (Supplementary Figure S10) that shows the timescale changes for each monkey individually. This new visualization, in conjunction with the statistics, confirms that while the magnitude of the effect differed between animals, the effect was consistent in direction and statistically significant in both. PPC inactivation led to a significant increase in intrinsic timescales for both Monkey J ($p = 3.02 \times 10^{-2}$, effect size = 0.19) and Monkey Q ($p = 4 \times 10^{-15}$, effect size = 0.63).

Rebuttal Figures

Rebuttal Fig. 1 | Noise correlations are strongest between pairs of long-timescale neurons. Boxplots compare the distribution of spike count noise correlations during the baseline period for three types of neuron pairs: short-timescale neurons (short-short), long-timescale neurons (long-long), and mixed pairs (long-short). Each box indicates the median (center line) and the interquartile range (25th–75th percentiles), with whiskers representing the data range and circles indicating outliers. A Kruskal-Wallis test showed a significant difference among the groups, and pairwise comparisons using Dunn's post-hoc test with Bonferroni correction revealed that long-long pairs had significantly higher noise correlations than both short-short and long-short pairs. (***, $p < 0.001$).

References

- Arcaro, M. J., Pinsk, M. A., Chen, J., & Kastner, S. (2018). Organizing principles of pulvino-cortical functional coupling in humans. *Nature Communications*, *9*(1), 5382. <https://doi.org/10.1038/s41467-018-07725-6>
- Aussel, A., Fiebelkorn, I. C., Kastner, S., Kopell, N. J., & Pittman-Polletta, B. R. (2023). Interacting rhythms enhance sensitivity of target detection in a fronto-parietal computational model of visual attention. *eLife*, *12*, e67684. <https://doi.org/10.7554/eLife.67684>
- Boshra, R., & Kastner, S. (2022). Attention control in the primate brain. *Current Opinion in Neurobiology*, *76*, 102605. <https://doi.org/10.1016/j.conb.2022.102605>
- Bruce, C. J., & Goldberg, M. E. (1985). Primate frontal eye fields. I. Single neurons discharging before saccades. *Journal of Neurophysiology*, *53*(3), 603–635. <https://doi.org/10.1152/jn.1985.53.3.603>
- Chaudhuri, R., Knoblauch, K., Gariel, M.-A., Kennedy, H., & Wang, X.-J. (2015). A Large-Scale Circuit Mechanism for Hierarchical Dynamical Processing in the Primate Cortex. *Neuron*, *88*(2), 419–431. <https://doi.org/10.1016/j.neuron.2015.09.008>
- Chen, X., Zirnsak, M., Vega, G. M., Govil, E., Lomber, S. G., & Moore, T. (2020). Parietal Cortex Regulates Visual Saliency and Saliency-Driven Behavior. *Neuron*, *106*(1), 177-187.e4. <https://doi.org/10.1016/j.neuron.2020.01.016>
- Fiebelkorn, I. C., & Kastner, S. (2019). A Rhythmic Theory of Attention. *Trends in Cognitive Sciences*, *23*(2), 87–101. <https://doi.org/10.1016/j.tics.2018.11.009>
- Hanes, D. P., & Schall, J. D. (1996). Neural Control of Voluntary Movement Initiation. *Science*, *274*(5286), 427–430. <https://doi.org/10.1126/science.274.5286.427>
- Litwin-Kumar, A., & Doiron, B. (2012). Slow dynamics and high variability in balanced cortical networks with clustered connections. *Nature Neuroscience*, *15*(11), 1498–1505. <https://doi.org/10.1038/nn.3220>
- Murray, J. D., Bernacchia, A., Freedman, D. J., Romo, R., Wallis, J. D., Cai, X., Padoa-Schioppa, C., Pasternak, T., Seo, H., Lee, D., & Wang, X.-J. (2014). A hierarchy of intrinsic timescales across primate cortex. *Nature Neuroscience*, *17*(12), 1661–1663. <https://doi.org/10.1038/nn.3862>
- Ponce-Alvarez, A. (2025). Network Mechanisms Underlying the Regional Diversity of Variance and Time Scales of the Brain's Spontaneous Activity Fluctuations. *Journal of Neuroscience*, *45*(10). <https://doi.org/10.1523/JNEUROSCI.1699-24.2024>

- Schall, J. D., Morel, A., King, D. J., & Bullier, J. (1995). Topography of visual cortex connections with frontal eye field in macaque: Convergence and segregation of processing streams. *The Journal of Neuroscience: The Official Journal of the Society for Neuroscience*, *15*(6), 4464–4487.
<https://doi.org/10.1523/JNEUROSCI.15-06-04464.1995>
- Shi, Y.-L., Zeraati, R., Laboratory, I. B., Levina, A., & Engel, T. A. (2025). Brain-wide organization of intrinsic timescales at single-neuron resolution (p. 2025.08.30.673281). bioRxiv.
<https://doi.org/10.1101/2025.08.30.673281>
- Stanton, G. B., Bruce, C. J., & Goldberg, M. E. (1995). Topography of projections to posterior cortical areas from the macaque frontal eye fields. *Journal of Comparative Neurology*, *353*(2), 291–305.
<https://doi.org/10.1002/cne.903530210>
- Wasmuht, D. F., Spaak, E., Buschman, T. J., Miller, E. K., & Stokes, M. G. (2018). Intrinsic neuronal dynamics predict distinct functional roles during working memory. *Nature Communications*, *9*(1), Article 1.
<https://doi.org/10.1038/s41467-018-05961-4>
- Xia, R., Chen, X., Engel, T. A., & Moore, T. (2024). Common and distinct neural mechanisms of attention. *Trends in Cognitive Sciences*, *28*(6), 554–567. <https://doi.org/10.1016/j.tics.2024.01.005>
- Zeisler, Z. R., Love, M., Rutishauser, U., Stoll, F. M., & Rudebeck, P. H. (2025). Consistent Hierarchies of Single-Neuron Timescales in Mice, Macaques, and Humans. *Journal of Neuroscience*, *45*(19).
<https://doi.org/10.1523/JNEUROSCI.2155-24.2025>

We thank the reviewers for their positive evaluations and for recommending the manuscript for publication. We have addressed the remaining minor suggestions as follows.

Black bold: Reviewer comment

Red regular: Our response

Reviewer Comments

Reviewer #1:

I have no further comments. The authors have satisfactorily addressed my previous concerns.

Reviewer #2:

The authors have satisfactorily addressed all my concerns, and I endorse the paper for publication. A minor note: panel B in Figure S2 seems unnecessary, as correlating half of the data with the full dataset will inherently produce a strong correlation.

We agree with the reviewer and have removed panel S2B from the Supplementary Information. Figure S2 now includes only the split-half correlation (first half vs. second half). We have updated the corresponding Supplementary Figure legend and Methods section accordingly.

Reviewer #3:

All concerns from my previous review are adequately addressed; I have no further comments, and can recommend the manuscript for publication.

Reviewer #4:

The revised version of the manuscript has been significantly improved, and the authors have convincingly addressed all my concerns.

More specifically:

- Regarding Point 1: The authors did not change the Fig. 6 but better refined the caption emphasizing in the Discussion that it serves only as a schematic illustration of how a broad fronto-parietal network can reciprocally modulate the timescales expressed by FEF neurons. This is a sufficient effort as I mainly asked to clarify that the inactivation of PPC can have widespread effects and only indirectly influence FEF. This point is now clearly stated in the Discussion.

- Regarding Point 2: The added text in the Discussion now clarifies the extent to which the relationship between neural activity and behavioral modulation is addressed in this work, thereby solving the issue of overinterpretation of the presented results.

- **Regarding Point 3: The authors added a new supplementary figure that convincingly addresses my concern. A minor point to be considered at this stage is to provide the parameters used for the kernel density estimates in the caption of Fig. S4 – for SUA (left) and MUA (right) – as they are likely different.**

We updated the Fig. S4 legend to report the kernel density bandwidth parameters. The legend now specifies that KDEs used R's default bandwidth and reports the resulting bandwidth values ($h = 16.00$ ms for SUA; $h = 12.29$ ms for MUA).

- Regarding Point 4: The authors have included the requested citation and added a clarifying text in the Discussion fully addressing my criticism.

- Regarding Point 5: The Supplementary Fig. S10 convincingly demonstrates that, despite my concerns, PPC inactivation increases intrinsic neural timescales in both monkeys.